# Preference-grounded Token-level Guidance for Language Model Fine-tuning

**Shentao Yang[1],   Shujian Zhang[1],   Congying Xia[2],   Yihao Feng[2],**
**Caiming Xiong[2],   Mingyuan Zhou[1]**
[1]The University of Texas at Austin        [2]Salesforce Research
shentao.yang@mccombs.utexas.edu,   yihao.ac@gmail.com
mingyuan.zhou@mccombs.utexas.edu

## Abstract

Aligning language models (LMs) with preferences is an important problem in natural language generation. A key challenge is that preferences are typically provided at the *sequence level* while LM training and generation both occur at the *token level*. There is, therefore, a *granularity mismatch* between the preference and the LM training losses, which may complicate the learning problem. In this paper, we address this issue by developing an alternate training process, where we iterate between grounding the sequence-level preference into token-level training guidance, and improving the LM with the learned guidance. For guidance learning, we design a framework that extends the pairwise-preference learning in imitation learning to both variable-length LM generation and the utilization of the preference among multiple generations. For LM training, based on the amount of supervised data, we present two *minimalist* learning objectives that utilize the learned guidance. In experiments, our method performs competitively on two distinct representative LM tasks — discrete-prompt generation and text summarization. Source codes are released at https://github.com/Shentao-YANG/Preference_Grounded_Guidance.

## 1   Introduction

Language models (LMs) have been successfully trained with token-level cross-entropy losses, where each token position has a corresponding term in the overall training losses [1–11]. Recent studies have shown that LMs can be further improved by aligning them with preferences from human feedback [12–15] or automatic evaluation metrics [16–18]. Typically, the preferences are only provided at the *sequence level*, *e.g.*, "Which of the two generated text sequences is better?" To align LMs with sequence-level preferences, there exist a variety of approaches, such as applying external filters to the training texts [19], performing supervised learning on some curated/improved datasets [20–22], and optimizing the LMs based on a learned sequence-level (pairwise-) preference predictor [14, 23–25].

While these approaches have contributed to the development of several revolutionary products [*e.g.*, 18, 15], a mismatch issue has emerged from the perspective of guiding LM fine-tuning. Concretely, the sequence-level preference is not grounded into the token level, where LM training losses occur. This means that there is a *mismatch* in granularity between the feedback and training losses — the preference is coarse-grained while the training losses are fine-grained. This issue is similar to the delayed-feedback problem in reinforcement learning (RL) [26–28], where informative feedback is available only at the end of the trajectory (sequence) and not at any of the intermediate timesteps. Previous studies have noted that this problem could have a negative impact on the empirical performance of the resulting LMs [29, 30], as it introduces a more challenging learning problem characterized by higher gradient variance and lower sample efficiency to achieve the learning goal [31, 32].

37th Conference on Neural Information Processing Systems (NeurIPS 2023).

To address this granularity mismatch, we focus on the following question: *How can we effectively ground sequence-level preference into token-level guidance for LM fine-tuning?* We propose an alternate training process that alternates between two stages: learning preference-grounded token-level guidance and improving the LM using the learned guidance. This alternate process reduces the requirement on supervised data and targets the low-data regime, *e.g.*, few/zero-shot learning, where task-specific supervised (pre-)training is infeasible and initial LMs have weak zero-shot abilities.

To ground the sequence-level preference into token-level guidance, we propose a framework for learning a token-level "reward" function[1], inspired by reward-learning-from-preferences in the imitation learning (IL) literature [33–35]. Specifically, we train the token-level rewards such that the corresponding evaluation for a generated text sequence reflects the preference among multiple alternative generations, where the preference comes from task-specific evaluation. While *summation* is classically used in IL to aggregate the learned token-level rewards into the text-sequence evaluation, LM tasks can be different from classical IL tasks. To cater to LM generations, our guidance-learning framework can accommodate more careful choices of the aggregation function beyond the classical summation. For instance, in generating text prompts to steer an LM for text classification, a "key token" in the prompt may be more effective than several mediocre tokens. Hence, using *maximum* to aggregate the token-level rewards may better reflect the text-sequence quality than *summation*.

To utilize the learned preference-grounded guidance in LM training, we present two *minimalist* learning objectives that contain only a minimal number of hyperparameters. These two objectives respectively target different amounts of supervised data in the specific LM task. We evaluate our framework on two distinct representative LM tasks: generating discrete text prompts for few-shot text classification and text summarization. On both tasks, our method exhibits competitive performance.

## 2   Main Method

Before diving into technical details, we will first establish the notation, provide some background on classical pairwise-preference learning, and offer an overview of our preference-grounding process.

**Notation.**  In most LM tasks, we are given a dataset $\mathcal{D} = \{(x^i, y^i)\}_{i=1}^N$ of $N$ supervised examples, where $x$ is the input to the LM, which can be a dummy, and $y$ is the target text-sequence. We denote the LM parameterized by $\theta$ as $\pi_\theta$. The $t^{\text{th}}$ generated token is denoted as $a_t$, given by $a_t \sim \pi_\theta(\cdot \mid s_t)$, where the context for token generation at step $t \geq 0$ is denoted as $s_t$, consisting of the LM input $x$ and the previously generated tokens $a_{<t} = (a_0, \ldots, a_{t-1})$. Specifically, $s_0 = x$ and $\forall\, t > 0, s_t = (x, a_{<t})$. The full generated text-sequence of length $T$ is denoted as $\boldsymbol{a} = (a_0, \ldots, a_{T-1})$. In most LM tasks, we have a task-specific evaluation metric $\mathcal{R}(s_T, y) \in \mathbb{R}$ that depends on the final context $s_T$ of the generated sequence and the target sequence $y$, with $s_T = (x, \boldsymbol{a})$. The objective of LM training is often to maximize the expected task-specific evaluation, which can be expressed as

$$\max_\theta \mathbb{E}_{(x,y)\sim\mathcal{D}} \mathbb{E}_{\boldsymbol{a}\sim\prod_{t=0}^{T-1} \pi_\theta(a_t \mid s_t)} \left[ \mathcal{R}(s_T = (x, \boldsymbol{a}), y) \right].$$

We model the learned token-level guidance as a bounded (reward) function $r_\phi(s_t, a_t) \in [0, 1]$, parametrized by $\phi$. Unlike the original sequence-level preference or evaluation that is only available at the final step $T$, the trained $r_\phi$ can densely guide the token selection at each step $t \leq T$.

**Pairwise Preference Learning.**  In reward-learning-from-preferences [*e.g.*, 33–35], a dense reward function is learned such that the *sum-aggregated* reward for the entire generation trajectory aligns with the pairwise preference between two trajectories. In the context of LM generation, suppose we have two text-generation trajectories $\tau^i$ and $\tau^j$ associated with the same LM input and target $(x, y)$, taking the form $\tau^i = \{(s_0^i, a_0^i), \ldots, (s_{T^i-1}^i, a_{T^i-1}^i)\}$ with sequence lengths $T^i$ and $T^j$, respectively. Assume that $\tau^i$ is preferred over $\tau^j$, denoted as $\tau^i \succ \tau^j$. A token-level reward function $r_\phi(s_t, a_t)$ is learned by requiring $\sum_{t=0}^{T^i-1} r_\phi(s_t^i, a_t^i) > \sum_{t=0}^{T^j-1} r_\phi(s_t^j, a_t^j)$. Following the Bradley-Terry model of preferences [36], the pairwise-preference loss for reward-function learning is

$$\ell(\phi) = -\log\left[ \exp\left( \sum_{t=0}^{T^i-1} r_\phi(s_t^i, a_t^i) \right) \Big/ \sum_{k\in\{i,j\}} \exp\left( \sum_{t=0}^{T^k-1} r_\phi(s_t^k, a_t^k) \right) \right], \tag{1}$$

which is often interpreted as binary classification in the literature [37–39]. In Eq. (1), *summation* $\sum(\cdot)$ is used to aggregate the learned token-level rewards into a parametrized sequence-level evaluation.

---

[1]We use the words "guidance" and "reward", "fine-tuning" and "training" interchangeably, depending on the specific context.

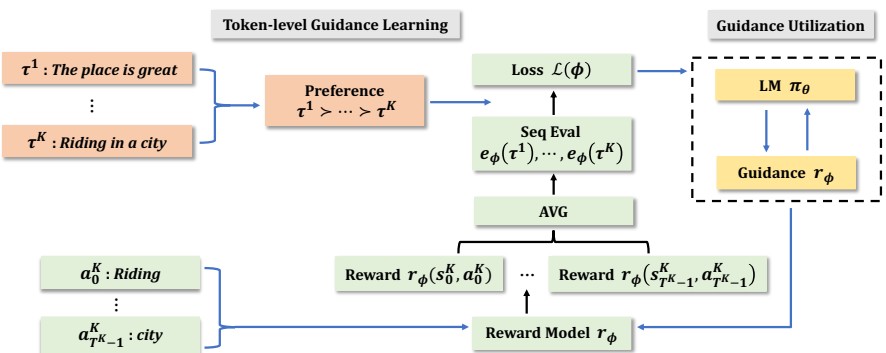

Figure 1: Overview of the proposed framework. "AVG" denotes *average*, which is an example of the aggregation function $f(\cdot)$ discussed in Section 2.1. "Seq Eval" refers to the parametrized sequence-level evaluations. The model choice of the reward function and LM depends on the specific task and is discussed in Section 4.

**Overview.** To ground the *sequence-level* preference into *token-level* guidance for LM training and thereby address the granularity mismatch discussed in Section 1, we present an alternate learning process that alternately learns the token-level guidance and trains the LM using the learned guidance.

For learning the preference-grounded guidance, in Section 2.1 we propose a framework that learns a token-level reward function that reflects the preference among multiple generated sequences. To utilize the learned preference-grounded guidance, based on the amount of supervised data in the specific task, in Section 2.2 we present two *minimalist* LM training approaches that require only minimal tuning. In our framework, we iterate between the above two steps to mitigate the distribution shift between the text sequences used to train the reward function and the text sequences evaluated by the reward function during LM training, taking into account that LMs can evolve during the training process. Our alternate-learning procedure is illustrated in Fig. 1.

## 2.1 Token-level Guidance Learning for Preference Grounding

Instead of applying the pairwise approach discussed above, we utilize the preference among multiple generated text sequences to learn the reward function $r_\phi(s_t, a_t)$. Intuitively, we use more information to train the reward function at each optimization step. Therefore, our approach can be more efficient and effective, especially when the optimization budget is limited.

Concretely, suppose we have $K \geq 2$ generated text-sequences $(\boldsymbol{a}^1, \ldots, \boldsymbol{a}^K)$ for the same LM input and target $(x, y)$, with the associated generation trajectories $(\tau^1, \ldots, \tau^K)$ and with possibly unequal sequence-lengths $(T^1, \ldots, T^K)$. Assume that there is a preference ordering among these $K$ sequences, where by "preference" we mean a ranking of text sequences based on some evaluations of full text-sequences. For description simplicity, let the preference ordering be $\boldsymbol{a}^1 \succ \cdots \succ \boldsymbol{a}^K \iff \tau^1 \succ \cdots \succ \tau^K$. We make no assumption about the source of preference. It may come from human ranking or some task-specific evaluation metric $\mathcal{R}$ on the full text-sequences, *e.g.*, the descending ordering of the text-sequence evaluations $\mathcal{R}(s_{T^1}^1, y) > \cdots > \mathcal{R}(s_{T^K}^K, y)$.

For a trajectory $\tau^k = \{(s_0^k, a_0^k), \ldots, (s_{T^k-1}^k, a_{T^k-1}^k)\}$, our desired token-level reward function $r_\phi$ generates a reward for each step as $\{r_\phi(s_t^k, a_t^k)\}_{t=0}^{T^k-1}$. A sequence-level evaluation $e_\phi(\tau^k)$ for trajectory $\tau^k$ can be obtained by $e_\phi(\tau^k) = f(\{r_\phi(s_t^k, a_t^k)\}_{t=0}^{T^k-1})$, where $f(\cdot)$ is the aggregation function over all per-step rewards, *e.g.*, the classical *summation* $\sum(\cdot)$. Our goal is to train $r_\phi$ such that these parametrized sequence-level evaluations $\{e_\phi(\tau^k)\}_{k=1}^K$ align with the given preference ordering $\tau^1 \succ \cdots \succ \tau^K$. Through this, the sequence-level preference is grounded into token-level rewards $r_\phi$.

Under the Plackett–Luce choice model [40, 41], the parametrized sequence evaluations $\{e_\phi(\tau^k)\}_{k=1}^K$ induce a probability distribution over all possible permutations of the integers $\{1, \ldots, K\}$. We want to maximize the likelihood of the given preference ordering $\mathrm{ord} = (1, \ldots, K)$, *i.e.*,

$$\min_\phi \mathcal{L}(\phi) =: -\log P\left(\mathrm{ord} \,|\, \{e_\phi(\tau^k)\}_{k=1}^K\right), \; P\left(\mathrm{ord} \,|\, \{e_\phi(\tau^k)\}_{k=1}^K\right) = \prod_{k=1}^K \left\{ \exp(e_\phi(\tau^k)) \middle/ \sum_{i=k}^K \exp(e_\phi(\tau^i)) \right\}. \quad (2)$$

---

**Algorithm 1** A learning routine for the preference-grounded token-level reward function $r_\phi$.

---

**Input:** The LM $\pi_\theta$, initialized reward $r_\phi$, aggregation function $f(\cdot)$, reward-training steps $M_{\text{rew}}$.
**for** iter $\in \{1, \ldots, M_{\text{rew}}\}$ **do**
    Use $\pi_\theta$ to generate $K$ sequences $\{\boldsymbol{a}^k\}_{k=1}^K$; and get the preference ordering among $\{\boldsymbol{a}^k\}_{k=1}^K$.
    With $f(\cdot)$, get sequence evaluations $\{e_\phi(\tau^k)\}_{k=1}^K$ from $r_\phi$; and optimize $r_\phi$ by Eq. (2).
**end for**

---

When $K = 2$ and $f(\cdot)$ denotes *summation*, Eq. (2) reduces to the classical pairwise-preference loss in Eq. (1). Therefore, our reward-learning loss can be viewed as an extension of the classical pairwise loss. Further, Eq. (2) extends the ListMLE loss [42] in recommender systems into preference learning under multiple variable-length trajectories.

Algo. 1 summarizes our reward-learning framework by describing an online-learning routine for training $r_\phi$. An offline or hybrid version can be obtained with minor changes.

**The Choice of Aggregation Function $f(\cdot)$.** In classical IL tasks such as robotics [43], the robots are trained to stand or walk as long as possible. In this scenario, *summation* is a natural choice for the aggregation function $f(\cdot)$. However, in many text generation tasks, such as summarization, the generation quality may not be directly associated with the length of the generated text sequence. Nevertheless, suppose the token-level rewards are positive (*i.e.*, $r_\phi > 0$), a longer sequence naturally has a higher sum of per-step rewards than a shorter one, which can bias $r_\phi$ towards automatically ranking longer sequences higher. This bias can hinder our reward-learning goal of aligning $\{e_\phi(\tau^k)\}_{k=1}^K$ with the given preference ordering. A naïve numeric example is additionally provided in Appendix C.

To mitigate the potential length bias in the classical *summation*, we discuss three alternative choices of the aggregation function $f(\cdot)$: *average*, *soft maximum*, and *soft minimum*.

***Average*.** We define the *average-aggregated* sequence-level evaluation $e_\phi^{\text{avg}}(\tau^k)$ for trajectory $\tau^k$ as

$$e_\phi^{\text{avg}}(\tau^k) = \tfrac{C}{T^k} \sum_{t=0}^{T^k-1} r_\phi(s_t^k, a_t^k), \quad C = \tfrac{1}{K} \sum_{k=1}^K T^k, \tag{3}$$

where $C$ is the average length of the $K$ sequences. Multiplied by the average length $C$ has the benefit of scaling $e_\phi^{\text{avg}}$ to the scale of $e_\phi^{\text{sum}}$, which ensures numerical-scale consistency with $e_\phi^{\text{sum}}$ and thus reduces hyperparameter tuning when switching from *summation* to *average* aggregation.

***Soft Maximum*.** We define the *soft-maximum-aggregated* sequence-level evaluation $e_\phi^{\max}(\tau^k)$ as

$$e_\phi^{\max}(\tau^k) = C \times \beta \cdot \log \left[ \sum_{t=0}^{T^k-1} \exp(r_\phi(s_t^k, a_t^k)/\beta) \right], \tag{4}$$

where $C$ is the average trajectory-length in Eq. (3) and $\beta$ is the temperature parameter.

***Soft Minimum*.** The *soft-minimum-aggregated* sequence-level evaluation $e_\phi^{\min}(\tau^k)$ follows Eq. (4) except for changing $\beta$ to $-\beta$.

## 2.2 LM Training with Preference-grounded Token-level Guidance

Considering the supervised-data availability, we present two *minimalist* LM training objectives that utilize the learned preference-grounded guidance: **1)** a REINFORCE-style update when there is no supervised data; **2)** reward-weighted MLE when there are sufficient data. Our LM training directly starts from raw pre-trained checkpoints, without task-specific supervised pre-training. This choice is to keep the algorithm general and consistent in both situations we consider, since task-specific pre-training may not be feasible in the setting of few/zero-shot learning.

As shown in Algo. 1, we train the reward function $r_\phi$ by the sequences sampled from LM $\pi_\theta$. Since task-specific pre-training to $\pi_\theta$ is not assumed, over the course of training, $\pi_\theta$ itself can evolve from a less-preferred distribution to a highly-preferred one. To mitigate the impact of this distribution shift and keep $r_\phi$ as accurate guidance for LM training, we periodically re-estimate $r_\phi$ during the first half of the LM-training process[2], motivated by recent works in model-based RL [44–47].

---

[2] In our preliminary study, we observed that this choice ("retraining the reward model only during the *first half* of the LM-training process") can save about 25-30% compute without hurting the performance much, compared to the vanilla reward retraining ("retraining the reward model throughout the *entire* LM-training process").

---

**Algorithm 2** An alternate-learning process for the reward function $r_\phi$ and the LM $\pi_\theta$.

---

**Input:** The dataset $\mathcal{D}$, initialized LM $\pi_\theta$, initialized reward function $r_\phi$, LM-training steps $M_{\mathrm{LM}}$, reward-retrain period $M_{\mathrm{re}}$, all inputs for training the reward function specified in Algo. 1.
**Initialize** $r_\phi$ by Algo. 1.
**for** iter $\in \{1, \ldots, M_{\mathrm{LM}}\}$ **do**
    **if** iter $\leq M_{\mathrm{LM}}/2$ and iter $\% M_{\mathrm{re}}$ == 0 **then**
        Re-train $r_\phi$ by Algo. 1 without re-initialization.
    **end if**
    Optimize $\pi_\theta$ by Eq. (5) or Eq. (6) with $\mathcal{D}$ and $r_\phi$.
**end for**

---

**Without Supervised Data.** When the LM $\pi_\theta$ needs to discover good text generations by itself, the learned token-level reward $r_\phi$ can be used to provide dense guidance on generating each token, *i.e.*, given the generation context $s_t$, select the next token $a_t$ such that $r_\phi(s_t, a_t)$ is high. Intuitively, for a generation trajectory $\tau$, if $\forall (s_t, a_t) \in \tau, r_\phi(s_t, a_t)$ is high, then the corresponding sequence-level evaluation $e_\phi(\tau) = f(\{r_\phi(s_t, a_t)\}_{t=0}^{T-1})$ can be also high, *e.g.*, the average or summation of token-level rewards. The associated text sequence $\boldsymbol{a}$ will thus be preferable since $r_\phi$ is trained to reflect the sequence-level preference (Section 2.1). Through $r_\phi$, the sequence-level preference is grounded into dense token-level guidance for LM training, without granularity mismatch or feedback delay.

With the learned $r_\phi$, a *minimalist* implementation of this LM-training idea is the discrete-variable optimization problem

$$\max_\theta \mathbb{E}_{t \sim \mathrm{Uniform}\{0, \ldots, T-1\}} \mathbb{E}_{a_t \sim \pi_\theta(\cdot \,|\, s_t)}[r_\phi(s_t, a_t)],$$

for each timestep $t$ of which we calculate its gradient by the classical REINFORCE method [48–50] since it can cope with a large vocabulary size. Here, $T$ denotes a generic sequence length. Additionally, since we want multiple text generations in typical LM tasks, instead of only one, we relax the convergence of the REINFORCE method by adding a standard max-entropy gradient, which can help capture multiple good behavior-modes [51–53]. Thus, the LM $\pi_\theta$ is trained by the gradient

$$\mathbb{E}_{t \sim \mathrm{Uniform}\{0, \ldots, T-1\}} \left\{ \mathbb{E}_{a_t \sim \pi_\theta(\cdot \,|\, s_t)}[r_\phi(s_t, a_t) \cdot \nabla_\theta \log \pi_\theta(a_t \,|\, s_t)] + \alpha \cdot \nabla_\theta \mathcal{H}(\pi_\theta(\cdot \,|\, s_t)) \right\}, \quad (5)$$

where $\mathcal{H}(\pi_\theta(\cdot \,|\, s_t))$ is the Shannon entropy of $\pi_\theta(\cdot \,|\, s_t)$ and $\alpha$ is a balancing coefficient.

**With Supervised Data.** With a labelled dataset $\mathcal{D} = \{(x^i, y^i)\}_{i=1}^N$ and with the learned preference-grounded guidance $r_\phi$, a *minimalist* enhancement of the classical MLE LM-training is the token-level weighted-MLE, where the per-token weight is given by the learned reward-function $r_\phi$. Our intention is to emphasize the important tokens in the given sequence $y$ and downweight the unimportant ones, where the token importance given by $r_\phi$ grounds the sequence-level preference. Intuitively, this weighting scheme can better utilize the LM capacity and the optimization budget, and may thus improve upon the vanilla supervised loss [54, 16]. Specifically, the LM $\pi_\theta$ is trained by

$$\min_\theta -\mathbb{E}_{(x,y) \sim \mathcal{D}} \left[ \sum_{t=0}^{|y|-1} w_t \cdot \log \pi_\theta(y_t \,|\, s_t) \right], \text{ with } s_t = (x, y_{<t}) \text{ and } w_t = \frac{r_\phi(s_t, y_t)}{\sum_{t'=0}^{|y|-1} r_\phi(s_{t'}, y_{t'})}, \quad (6)$$

where $|y|$ is the length of the target sequence $y$ and $w_t$ is the self-normalized token-level reward. The standard self-normalization is used to reduce the gradient variance among the samples in $\mathcal{D}$.

Algo. 2 sums up the entire alternate-learning process, with the reward-learning routine in Algo. 1.

## 3 Related Work

**Guiding Signals for LM Training.** One string of works in LM training directly optimizes the LMs against the native sequence-level feedback such as the test-time metric [*e.g.*, 3, 55–59, 32]. This choice, however, may directly suffer from the delayed-feedback issue discussed in Section 1 and the subsequent high gradient variance and low sample efficiency [31, 32]. In the recent trend of RL-based LM training, it has been common to incorporate a token-level KL penalty towards the uniform distribution [31, 60], the initial LM [23, 61], the supervised-fine-tuned model [12, 62–64, 14], or the base momentum model [65], to add to the delayed/ungrounded feedback. Although that KL penalty does impact the RL-based LM training at the token level, it is not tailored to the concrete task or the desired sequence-level feedback. When combined with the delayed-feedback issue, it

could distract the LM training from improving the received feedback/evaluation, especially at the beginning of the text-sequence generation, which can however affect all subsequent token selections. By contrast, as seen in Eq. (5), even when added a max-entropy gradient, our preference-grounded token-level guidance can still provide dense, task-specific, and feedback-oriented guidance on the selection of each token. For a more detailed discussion on the RL formulation of LM generation, the delayed-feedback issue in RL-based LM training, and delayed feedback with KL penalty, please refer to Appendix F.

In some relatively "ideal" settings, prior works have attempted to learn task-specific token-level guidance for LM training. For instance, Shi et al. [66] use inverse RL, Guo et al. [67] propose a hierarchical approach, and Yang et al. [68] learn LM discriminators; but these methods require abundant expert data for supervised (pre-)training, making them infeasible for the few/zero-shot settings we consider. Under the same requirement of sufficient expert data, Lin et al. [69] learn a sequence-level adversarial-ranking reward and Yu et al. [70] train a GAN structure. They both use Monte-Carlo rollouts to simulate intermediate rewards, which can be computationally expensive and have high variance. Le et al. [71] use some values related to the sequence evaluation without explicitly learning per-token rewards. Pang et al. [72] learn a token-level error predictor for machine translation, but they rely on expert error-span annotations for each translation, which is highly demanding.

By contrast, we propose a versatile framework for learning task-specific token-level guidance for LM training that can ground the sequence-level preference. Our approach is not limited to standard LM tasks and is also suitable for the low-data regime, with few assumptions about expert-data availability or preference source. In our experiments, we compare our method to recent RL-based approaches that train LM under delayed/ungrounded feedback with KL penalty. We discuss additional related works on prompt generation, text summarization, and aligning LMs with preferences in Appendix E.

## 4 Experiments

We test our framework on two distinct representative text-sequence generation tasks: **1)** input-agnostic discrete text-prompt generation for few-shot text-classification (Section 4.1), **2)** the classical text summarization (Section 4.2). Our LM training directly starts from raw pre-trained checkpoints from HuggingFace [73], without task-specific supervised pre-training. Depending on the LM $\pi_\theta$ used in the specific task, our reward function $r_\phi$ can be implemented as either a decoder-only or an encoder-decoder model. Similar to prior works [*e.g.*, 32, 25], given a text sequence $\boldsymbol{a}$ and an LM input $x$, the causal mask in transformers enables us to get the learned guidance $r_\phi(s_t, a_t)$ at each step of the sequence in parallel. Source codes have been publicly released.

### 4.1 Input-agnostic Discrete-prompt Generation

**Overview.** In discrete text-prompt generation [*e.g.*, 10, 74], we input a discrete text-prompt $\boldsymbol{a}$ and an observation sequence $o$ to a large pre-trained downstream LM $\pi_{\mathrm{DLM}}(\cdot \mid \boldsymbol{a}, o)$ to directly classify text $o$, without finetuning $\pi_{\mathrm{DLM}}$. We follow the classical setting [*e.g.*, 75, 60] to perform classification by selecting tokens corresponding to some predefined class labels. In our input-agnostic setting, the generated prompt is independent of the observation $o$. During inference time, only the learned prompts are used and the LM $\pi_\theta$ is discarded. The initial input $x$ to $\pi_\theta$ is a dummy, and the target $y$ is the class label. We also adopt the standard few-shot setting [76], where both the training and validation sets have $16\,(o, y)$-pairs per class. With a fixed length $T$, the goal is to find discrete text-prompts $\boldsymbol{a} = (a_0, \ldots, a_{T-1})$ that have high test accuracy. We simulate the sequence-level preference by the stepwise metric in Deng et al. [60], *i.e.*, the higher value the better prompt. This choice ensures a fair comparison and avoids a potential overfitting — training and testing the LM on the same evaluation metric "accuracy". Appendix D discusses more details about the prompt task.

**LM Training, Implementation, and Datasets.** Since the prompt-generation task does not assume the availability of supervised data — the ground-truth prompts, the LM $\pi_\theta$ is trained by the REINFORCE-style update in Section 2.2 to discover highly-accurate prompts by itself. We implement our framework on the codebase of RLPrompt [60], and adopt the standard datasets and most hyperparameter settings in it. Reward training is reconducted every 1000 steps during the first 6000 steps of the LM training process and has early stopping. Reward function is learned with 5 sampled sequences and the temperature in Eq. (4) is set as $\beta = 2$. The coefficient $\alpha$ in Eq. (5) is $\alpha = 2^{-3}$. Appendix A.2 discusses the choices of these hyperparameters. The length of the generated prompts is fixed at 5. We

Table 1: Test accuracy on the prompt task. Best overall result is bold and best discrete-prompt result is underlined if different. The reported results are mean (standard deviation). We denote "BB Tuning-50" for Black-Box Tuning with mixed discrete and soft prompts that tunes the 50 soft tokens; and "AVG", "SUM", "MIN", "MAX" for our method with aggregation function average, summation, soft minimum, and soft maximum (Section 2.1).

| | | SST-2 | Yelp P. | AG News |
|---|---|---|---|---|
| Finetuning | Few-shot Finetuning | 80.6 (3.9) | 88.7 (4.7) | **84.9** (3.6) |
| Continuous Prompt | Soft Prompt Tuning [83] | 73.8 (10.9) | 88.6 (2.1) | 82.6 (0.9) |
| | BB Tuning-50 [78] | 89.1 (0.9) | 93.2 (0.5) | 83.5 (0.9) |
| | AutoPrompt [84] | 75.0 (7.6) | 79.8 (8.3) | 65.7 (1.9) |
| Discrete Prompt | Manual Prompt [85] | 82.8 | 83.0 | 76.9 |
| | In-Context Demo [10] | 85.9 (0.7) | 89.6 (0.4) | 74.9 (0.8) |
| | Instructions [86] | 89.0 | 84.4 | 54.8 |
| | GrIPS [87] | 87.1 (1.5) | 88.2 (0.1) | 65.4 (9.8) |
| | RLPrompt [60] | 90.5 (1.5) | 94.2 (0.7) | 79.7 (2.1) |
| | Ours (AVG / SUM) | **92.6** (1.7) | 94.7 (0.6) | 82.8 (1.5) |
| | Ours (MIN) | 91.9 (1.8) | 94.4 (0.8) | 82.4 (1.1) |
| | Ours (MAX) | 91.2 (2.5) | **94.8** (0.5) | 83.3 (1.4) |

test on three popular few-shot datasets in prior work [*e.g.*, 77, 78]: two sentiment binary-classification datasets SST-2 [79, 80] and Yelp Polarity [81], and a topic four-way-classification dataset AG News [81, 82]. Additional details on the experiment and datasets are provided in Appendix B.1.

**Results.** We compare three variants of our framework with finetuning and with baselines in discrete- and continuous-prompt generation. Since the generated prompts all have length 5, in this task, the *average* aggregation is equivalent to *summation*. Table 1 shows the test accuracy, where we rerun the codebase of RLPrompt [60] under the same random seeds and evaluation script as our method.[3] Other baseline results are from the literature [60, 88].

On all three tested datasets, our method shows competitive and stable results against the strong baselines not only in discrete-prompt generation, but also in heavier continuous-prompt tuning and finetuning the large downstream LM. Based on Section 3, the performance improvement achieved by our method compared to RLPrompt suggests that utilizing the token-level guidance learned by our approach, which grounds the task-specific preference, can be more effective than learning under delayed/ungrounded feedback with KL penalty. Further, on both Yelp P. and AG News, using MAX aggregation is better than the classical *summation*. Table 3 in Appendix A shows examples of good generated prompts and their test accuracy. For instance, high-quality prompts on the AG News dataset often contain a topic classification keyword, such as "Tags" and "Category". This aligns with our intuition that good prompts may be identified by a (few) "key" token(s), as discussed in Sections 1 and 2.1. Thus, the *(soft-)maximum* aggregation may better reflect prompt quality than *summation*.

## 4.2 Text Summarization

**Overview.** In the summarization task, we follow the standard setting [*e.g.*, 89, 61], where a set of supervised samples is available. The LM input $x$ is the text to be summarized and the target $y$ is the given summary. We simulate the sequence-level preference by the classical Meteor score [90] and report the standard ROUGE scores [91], to avoid overfitting evaluation metrics as in the prompt task.

**LM Training, Implementation, and Datasets.** Since a supervised dataset $\mathcal{D}$ is available in this task, the LM $\pi_\theta$ can be trained by the weighted-MLE objective in Section 2.2. This objective could be more stable and computationally efficient than REINFORCE-style methods in tasks of long-sequence generation. Due to limited computing resources, unless explicitly mentioned, we use the standard T5-small model [89] for both the LM and reward function. The reward training is simply 1 epoch of training on randomly sampled 10% of the training set and is repeated every 0.5 epochs during the first 2 epochs of LM training. Reward function is learned with 3 sampled sequences and again the temperature $\beta = 2$ in Eq. (4). Additional experiment details are in Appendix B.2. We test on the standard setting of two news summary datasets: CNN/DailyMail (CNN/DM) [92] and XSum [93].

**Results.** We compare four variants in our framework with the standard supervised fine-tuning and RL-based methods PPO and NLPO in RL4LMs [61] under the environmental reward Meteor — both

---

[3]There are small discrepancies between our reported RLPrompt results and the original paper's. We have confirmed our reproduced results both with RLPrompt's authors and with the recent TEMPERA paper [88].

Table 2: Results on text summarization. We bold the best result of each metric on each dataset. The results of Lead-3 on CNN/DM are from Ramamurthy et al. [61] and on XSum are from Lewis et al. [7]. Other baseline results are from our reruning RL4LMs' codebase [61] using T5-small. Number reporting formats follow Table 1.

| | CNN/DailyMail | | | XSum | | |
| | ROUGE-1 | ROUGE-2 | ROUGE-L | ROUGE-1 | ROUGE-2 | ROUGE-L |
| --- | --- | --- | --- | --- | --- | --- |
| Lead-3 | 40.10 | 17.50 | 36.30 | 16.30 | 1.60 | 11.95 |
| Supervised | 38.88 (0.02) | 16.22 (0.05) | 32.58 (0.04) | 31.79 (0.02) | 9.68 (0.01) | 24.70 (0.03) |
| PPO | 39.16 (0.51) | 17.37 (0.33) | 33.77 (0.37) | 23.18 (0.31) | 4.46 (0.19) | 16.07 (0.32) |
| Supervised + PPO | 39.17 (0.65) | 17.29 (0.44) | 33.76 (0.53) | 28.24 (0.39) | 7.68 (0.13) | 20.02 (0.23) |
| NLPO | 38.90 (0.35) | 17.22 (0.35) | 33.51 (0.42) | 22.97 (0.23) | 4.53 (0.13) | 15.62 (0.35) |
| Supervised + NLPO | 39.27 (0.60) | 17.41 (0.36) | 33.85 (0.42) | 28.08 (0.16) | 7.68 (0.20) | 19.88 (0.16) |
| Ours (AVG) | **40.94** (0.02) | **18.78** (0.03) | **38.17** (0.03) | **33.62** (0.03) | **11.17** (0.02) | **26.33** (0.05) |
| Ours (SUM) | 40.70 (0.06) | 18.48 (0.05) | 37.93 (0.08) | 33.27 (0.09) | 10.83 (0.07) | 25.90 (0.06) |
| Ours (MIN) | 40.78 (0.06) | 18.67 (0.03) | 38.01 (0.04) | 33.57 (0.02) | 11.14 (0.02) | 26.30 (0.03) |
| Ours (MAX) | 39.98 (0.08) | 18.06 (0.03) | 37.26 (0.06) | 32.50 (0.14) | 10.46 (0.12) | 25.58 (0.12) |

with and without task-specific supervised pre-training. For a fair comparison, the baseline results are from our rerunning RL4LMs' codebase with a T5-small model as our method.[4] Table 2 shows the mean and standard deviation of ROUGE-1/2/L score across three random seeds.

On both datasets, our method shows favorable and stable performance against the classical and recent baselines. The better results of our method over supervised fine-tuning confirm the improvement of our reward-weighted MLE over the vanilla supervised loss, as discussed in Section 2.2. As in the prompt task, the gain of our method over RL-based baselines may indicate the benefit of utilizing our preference-grounded token-level guidance over learning under delayed feedback with KL penalty. In this task, using *average* as the aggregation function outperforms the classical *summation*. This confirms our idea in Section 2.1 on avoiding the interference of unequal sequence-lengths in training $r_\phi$. Using MIN is also suitable for this task, since it is not confounded by individual lengths and reflects overall text quality. Unlike the prompt task, using MAX is unsuitable, since good summaries can hardly be identified by a few keywords. Overall, these results show the importance of customizing the aggregation choice for the specific LM task, a key feature of our guidance-learning framework.

Further, to verify the performance of our method under a larger LM, we change the *average* variant of our method in Table 2 from T5-small to using T5-base LM. Fig. 2 **(a)** – **(e)** compares our method on CNN/DM against the baselines, with an additional metric BertScore [94]. The baseline results are directly cited from RL4LMs [61] and are the per-metric best across their three environmental rewards.[5] Table 4 in Appendix A.1 shows the detailed numbers. It is clear that our method performs favorably against these strong baseline methods, especially in the ROUGE-L, BERTScore, Meteor, and ROUGE-2 metrics. To further varify our method, we conducted a human study under the T5-base LM. The results are in Fig. 2 **(f)**, with detailed setup and numerics in Table 5 of Appendix A.1. It is clear that this human evaluation on the summarization task supports the improvements in ROUGE, Meteor, and BertScore by our method. Further scaling-up of our method is left as future work.

## 4.3 Ablation Study

This section discusses the following three research questions to better understand our framework.

**(a):** *What will be the performance if we switch to using preference-based sequence-level guidance?*

To further study the gain of grounding preference into token-level guidance, we change the preference-based *token-level* reward in our method to the corresponding *sequence-level* reward. Fig. 3 shows the results when applying this change to our best variants in the prompt and summarization tasks in Sections 4.1 and 4.2, including the T5-base LM in Section 4.2, in comparison to the best corresponding baselines. For summarization, we plot the average ROUGE scores, *i.e.*, (ROUGE-1 + ROUGE-2 + ROUGE-L) / 3. Table 6 in Appendix A.1 shows each ROUGE metric with standard deviation.

We see that learning and using preference-based sequence-level guidance does not provide a significant advantage over those baselines that mostly directly work with the task-specific native sequence-level feedback — the results are even much worse than the baselines in some datasets. Besides, the results of our sequence-level variants are generally less stable. These echo the harm of the delayed-feedback issue discussed in Section 1. Overall, this set of comparisons confirms that the gain of our framework

---

[4]We carefully tuned the RL4LMs' baselines on several hyperparameters, which is detailed in Appendix B.2.
[5]The "ROUGE-L" here refers to "Rouge-LSum" in RL4LMs and HuggingFace, as detailed in Appendix B.2.

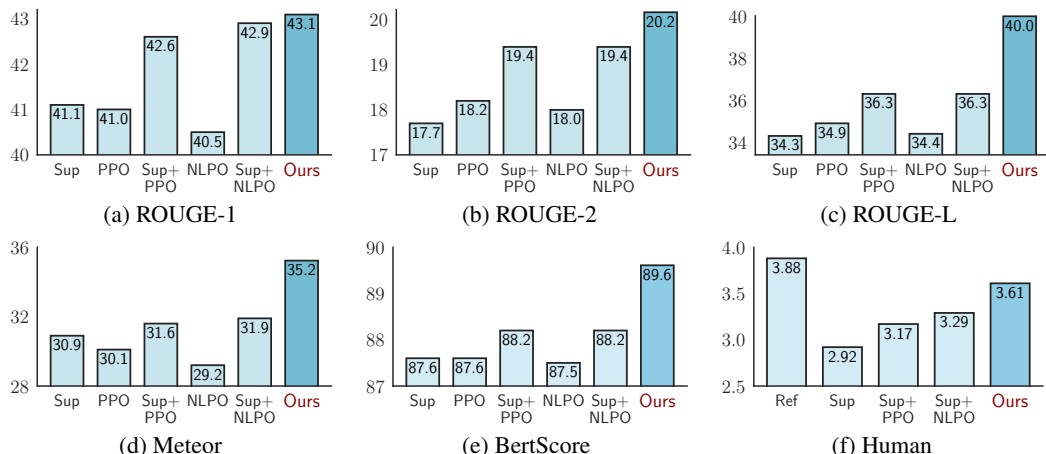

Figure 2: CNN/DM summarization of our method and baselines under **T5-base** LM. "Sup" denotes "Supervised". "Ref" denotes the ground-truth reference summary. Except for the human study in **(f)**, baseline results are directly cited from RL4LMs [61] and are the per-metric best across their three environmental rewards.

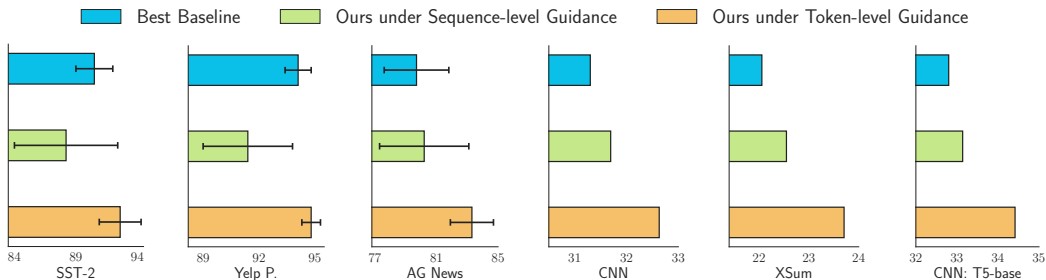

Figure 3: Performance of our method using sequence-level and token-level preference-based guidance. "Best Baseline" refers to the best result in the baseline discrete-prompt methods for the prompt task, and the best result over all baseline methods for the summarization task. Error bars show one standard deviation.

mainly comes from our preference-grounding perspective, *i.e.*, learning and using a preference-based *token-level* guidance, rather than simply learning and using "a preference-based guidance."

**(b):** *How does our method perform if we remove the reward-function retraining scheme?*

To study the effect of guidance re-estimation, we remove the reward-function retraining scheme from our best variants in the prompt and summarization tasks in Sections 4.1 and 4.2, including the T5-base LM in Section 4.2. Fig. 4 compares our methods with the best corresponding baselines. For the summarization task, we again plot the average ROUGE scores. Table 7 in Appendix A.1 shows each ROUGE metric with standard deviation. Appendix G discusses more on this re-estimation scheme.

Without guidance re-estimation, our method still performs competitively against the strong baselines, which corroborates the benefit of our preference-grounded guidance. Fig. 4 also verifies our intuition in Section 2 that the gain of this scheme depends on the zero-shot ability of the initial LMs. Specifically, in the prompt task where the initial LM has little zero-shot ability, reward-function retraining is helpful to both improve performance and reduce variance. In the summarization task where the initial LM does have some zero-shot ability (as shown in Ramamurthy et al. [61]), guidance re-estimation indeed helps results not as much, since the distribution-shift issue in Section 2 is less significant in this case. In this task, both our variants, with and without reward retraining, outperform the baselines.

**(c):** *What if we learn the token-level guidance by a different number of text sequences?*

To study how the number of sequences used to learn the reward function impacts our method's performance, we vary this number in the AVG variant in Tables 1 and 2. Fig. 5 shows the prompt results on SST-2 and summarization results on CNN/DM and XSum. For the latter, we again plot the average ROUGE scores. The scores of each ROUGE metric are in Tables 8 and 9 of Appendix A.1.

Recall that the best baseline result on SST-2 in Table 1 is 90.5, on CNN/DM and XSum in Table 2 is respectively 31.3 and 22.06. Thus, our method is generally robust to the number of sequences

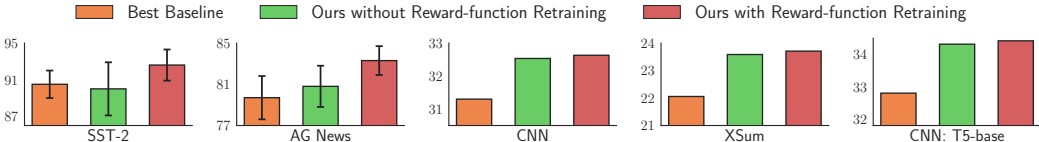

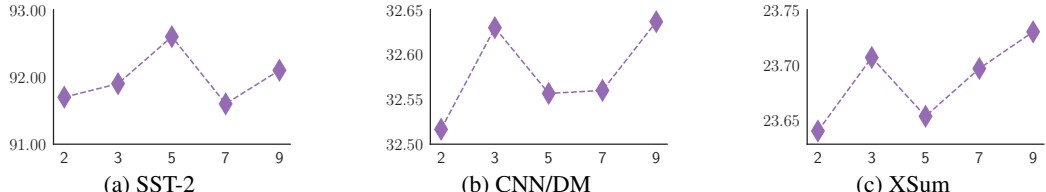

Figure 4: Performance of our method with and without the reward-function retraining scheme. "Best Baseline" refers to the same as in Fig. 3. Error bars show one standard deviation.

Figure 5: Varying the number of sequences to learn the token-level guidance, showing mean over random seeds.

used to learn the guidance. Compared with the classical pairwise-preference learning (Section 2), our framework has the flexibility in using multiple sequences. As illustrated in Fig. 5, using three or more sequences to learn the reward function can be generally more beneficial than using only two.

Due to the page limit, we defer additional ablation study to Appendix A.2, where we **(1)** show that our framework is generally robust to the hyperparameter $\beta$ in Eq. (4) and $\alpha$ in Eq. (5); **(2)** further validate the harm of the delayed-feedback issue to the relevant LM-training methods on longer text-sequence generation; **(3)** show that the efficacy of our framework is not tied to the specific preference sources considered in this section.

## 5    Conclusion

To address the granularity mismatch between the sequence-level preference and the token-level LM training losses, in this paper, we develop an alternate-learning process, where we iterate between grounding sequence-level preference into token-level training guidance, and training the LM with the learned guidance. Our method performs competitively on two distinct representative LM tasks. Future work includes combining our preference-grounded guidance with RL-based LM training, and applying our method to human preference and/or other tasks such as (task-oriented) dialog systems.

## Acknowledgments and Disclosure of Funding

S. Yang, S. Zhang, and M. Zhou acknowledge the support of NSF-IIS 2212418, NIH-R37 CA271186, the Texas Advanced Computing Center (TACC), and the NSF AI Institute for Foundations of Machine Learning (IFML). S. Yang acknowledges the support of the University Graduate Continuing Fellowship from UT Austin.

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

# Appendix for "Preference-grounded Token-level Guidance for Language Model Fine-tuning"

## Table of Contents

# A Additional Experimental Results

## A.1 Tabular Results

Table 3: Examples of the generated discrete input-agnostic text-prompt and their classification accuracy on the corresponding test set.

| SST-2 | | AG News | |
|---|---|---|---|
| Prompt | Accuracy | Prompt | Accuracy |
| guys filmmaker filmmaker rated Grade | 94.18 | newsIntroduction Comments Tags Search | 85.78 |
| MovieMovieFilm rated Grade | 94.18 | newsTopic Blog Support Category | 85.55 |
| Rated CinemaScoreReporting Grade | 94.01 | news RecentRecentPhotosIntroduction | 84.53 |
| employment theater rated Oscars Grade | 93.96 | news Recent Brief LatestExample | 84.51 |
| scene filmmaking rated comedian Grade | 93.85 | newsVirtualBlogBlogNet | 84.33 |

Table 4: Detailed results on CNN/DM summarization under **T5-base LM** for Section 4.2. We bold the best result of each metric. Baseline results are directly cited from RL4LMs [61]. "Env. Reward" denotes the environmental reward in RL4LMs. The "ROUGE-L" here refers to "Rouge-LSum" in RL4LMs and in the Hugging Face interface, which is discussed in details in Appendix B.2. In Section 4.2, we plot the results of our method with the *average* aggregation, which is the best variant in Table 2. We report the mean (standard deviation) of our method over three random seeds.

| Algorithm | Env. Reward | ROUGE-1 | ROUGE-2 | ROUGE-L | Meteor | BertScore |
|---|---|---|---|---|---|---|
| Lead-3 | | 40.1 | 17.5 | 36.3 | 33.3 | 87.4 |
| Supervised | | 41.1 | 17.7 | 34.3 | 30.9 | 87.6 |
| PPO | Rouge-1 | 41.0 | 18.2 | 34.9 | 27.6 | 87.6 |
| | Rouge-Avg | 39.6 | 17.6 | 33.8 | 27.0 | 87.4 |
| | Meteor | 40.8 | 17.8 | 34.2 | 30.1 | 87.3 |
| NLPO | Rouge-1 | 40.4 | 18.0 | 34.4 | 27.5 | 87.5 |
| | Rouge-Avg | 40.4 | 17.7 | 34.4 | 27.4 | 87.4 |
| | Meteor | 40.5 | 18.0 | 34.3 | 29.2 | 87.2 |
| Supervised + PPO | Rouge-1 | 41.7 | 18.9 | 35.8 | 27.8 | 88.2 |
| | Rouge-Avg | 42.5 | 19.4 | 36.3 | 29.6 | 88.2 |
| | Meteor | 42.6 | 19.4 | 36.1 | 31.6 | 88.0 |
| Supervised + NLPO | Rouge-1 | 42.1 | 19.3 | 36.1 | 28.7 | 88.2 |
| | Rouge-Avg | 42.4 | 19.3 | 36.3 | 29.5 | 88.2 |
| | Meteor | 42.9 | 19.4 | 36.1 | 31.9 | 88.0 |
| Ours (AVG) | | **43.09** (0.06) | **20.17** (0.04) | **39.99** (0.07) | **35.23** (0.06) | **89.61** (0.12) |
| Ours (SUM) | | 42.86 (0.08) | 19.92 (0.08) | 39.76 (0.11) | 34.74 (0.37) | 89.24 (0.11) |
| Ours (MIN) | | 42.92 (0.14) | 20.01 (0.02) | 39.84 (0.08) | 34.88 (0.13) | 89.33 (0.07) |
| Ours (MAX) | | 42.38 (0.17) | 19.49 (0.02) | 39.34 (0.09) | 34.13 (0.32) | 89.09 (0.19) |

**Setup and results of the human evaluation.** Table 5 below presents the results of our human evaluation on CNN/DM summarization under the T5-base LM. We generally adopt the protocol in Stiennon et al. [12] to evaluate the overall summary quality. Our model is compared with the baselines Supervised, Supervised+PPO, and Supervised+NLPO in RL4LMs [61]. The result of the reference summaries is also presented, which is intended for sanity check rather than method comparison. In conducting this evaluation, we randomly picked 100 articles in the test split of CNN/DM and showed to 20 qualified evaluators the summaries generated from each method, along with the article. The method names were anonymized. The evaluators were asked to read the article and score each summary. Summaries are scored on a 5-Point Likert Scale {1, 2, 3, 4, 5}, where score 5 is the highest and 1 the lowest. From Table 5, it is clear that human evaluation supports the improvements in ROUGE, Meteor, and BertScore by our method in Table 4.

Table 5: Average human ratings on CNN/DM summarization under the T5-base LM. We bold the best result apart from the ground-truth Reference summary. A detailed description on the setup is in the above text.

| | Supervised | Supervised+PPO | Supervised+NLPO | Ours | Reference |
|---|---|---|---|---|---|
| Average Human Rating | 2.92 | 3.17 | 3.29 | **3.61** | 3.88 |

Table 6: Scores on each ROUGE metric for our method using sequence-level and token-level preference-based guidance in the summarization tasks in Section 4.3 (**a**). "Seq." denotes our method with sequence-level preference-based guidance, and "Token" denotes our method with token-level preference-based guidance. The reported numbers are mean (standard deviation) over three random seeds. The row "Average" shows the average of the three ROUGE scores, *i.e.*, (ROUGE-1 + ROUGE-2 + ROUGE-L) / 3.

| | CNN/DM | | XSum | | CNN/DM (T5-base LM) | |
| | Seq. | Token | Seq. | Token | Seq. | Token |
|---|---|---|---|---|---|---|
| ROUGE-1 | 40.20 (0.07) | 40.94 (0.02) | 32.56 (0.08) | 33.62 (0.03) | 42.10 (0.15) | 43.09 (0.06) |
| ROUGE-2 | 17.80 (0.08) | 18.78 (0.03) | 9.98 (0.04) | 11.17 (0.02) | 19.23 (0.11) | 20.17 (0.04) |
| ROUGE-L | 37.08 (0.06) | 38.17 (0.03) | 25.11 (0.07) | 26.33 (0.05) | 38.09 (0.14) | 39.99 (0.07) |
| Average | 31.69 | 32.63 | 22.55 | 23.71 | 33.14 | 34.42 |

Table 7: Scores on each ROUGE metric for our method with and without the reward-function retraining scheme in the summarization tasks in Section 4.3 (**b**). "Without Retrain" denotes our method without reward-function retraining, and "With Retrain" denotes our method with reward-function retraining. The reported numbers are mean (standard deviation) over three random seeds. The row "Average" shows the average of the three ROUGE scores, *i.e.*, (ROUGE-1 + ROUGE-2 + ROUGE-L) / 3.

| | CNN/DM | | XSum | | CNN/DM (T5-base LM) | |
| | Without Retrain | With Retrain | Without Retrain | With Retrain | Without Retrain | With Retrain |
|---|---|---|---|---|---|---|
| ROUGE-1 | 40.83 (0.10) | 40.94 (0.02) | 33.45 (0.11) | 33.62 (0.03) | 42.98 (0.08) | 43.09 (0.06) |
| ROUGE-2 | 18.70 (0.07) | 18.78 (0.03) | 11.07 (0.06) | 11.17 (0.02) | 20.09 (0.06) | 20.17 (0.04) |
| ROUGE-L | 38.07 (0.09) | 38.17 (0.03) | 26.23 (0.10) | 26.33 (0.05) | 39.87 (0.08) | 39.99 (0.07) |
| Average | 32.53 | 32.63 | 23.58 | 23.71 | 34.31 | 34.42 |

Table 8: Scores on each ROUGE metric for the summarization task on CNN/DM in Section 4.3 (**c**), where we vary the number of sequences used to learn the token-level guidance. The reported numbers are mean (standard deviation) over three random seeds. The row "Average" shows the average of the three ROUGE scores, *i.e.*, (ROUGE-1 + ROUGE-2 + ROUGE-L) / 3.

| | Number of Sequences | | | | |
| | 2 | 3 | 5 | 7 | 9 |
|---|---|---|---|---|---|
| ROUGE-1 | 40.80 (0.06) | 40.94 (0.02) | 40.87 (0.09) | 40.86 (0.08) | 40.95 (0.01) |
| ROUGE-2 | 18.70 (0.04) | 18.78 (0.03) | 18.71 (0.02) | 18.74 (0.06) | 18.78 (0.01) |
| ROUGE-L | 38.05 (0.03) | 38.17 (0.03) | 38.09 (0.07) | 38.08 (0.08) | 38.18 (0.02) |
| Average | 32.52 | 32.63 | 32.56 | 32.56 | 32.64 |

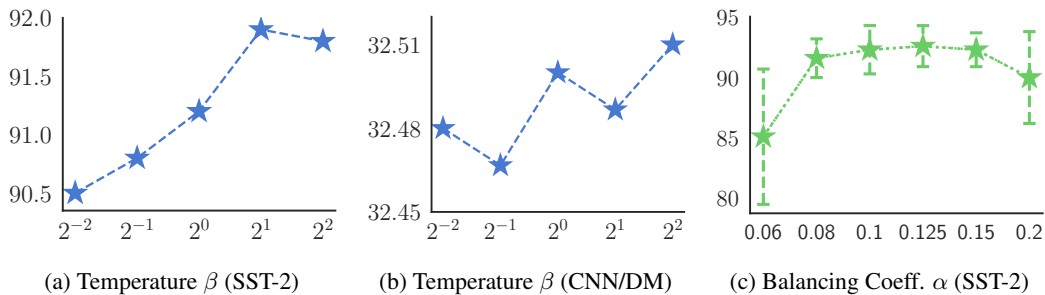

(a) Temperature $\beta$ (SST-2)  (b) Temperature $\beta$ (CNN/DM)  (c) Balancing Coeff. $\alpha$ (SST-2)

Figure 6: Line plots comparing the performance under different values of the hyperparameter $\beta$ in Eq. (4) and $\alpha$ in Eq. (5). The plotted numbers are mean over three random seeds. Error bars show one standard deviation.

Table 9: Scores on each ROUGE metric for the summarization task on XSum in Section 4.3 **(c)**, where we vary the number of sequences used to learn the token-level guidance. The reported numbers are mean (standard deviation) over three random seeds. The row "Average" shows the average of the three ROUGE scores, *i.e.*, (ROUGE-1 + ROUGE-2 + ROUGE-L) / 3.

| | Number of Sequences | | | | |
|---|---|---|---|---|---|
| | 2 | 3 | 5 | 7 | 9 |
| ROUGE-1 | 33.54 (0.06) | 33.62 (0.03) | 33.56 (0.08) | 33.56 (0.02) | 33.63 (0.02) |
| ROUGE-2 | 11.12 (0.04) | 11.17 (0.02) | 11.12 (0.05) | 11.19 (0.05) | 11.20 (0.03) |
| ROUGE-L | 26.26 (0.06) | 26.33 (0.05) | 26.28 (0.06) | 26.34 (0.06) | 26.36 (0.03) |
| Average | 23.64 | 23.71 | 23.65 | 23.70 | 23.73 |

## A.2 Further Ablation Study

In learning the preference-based *sequence-level* guidance in Section 4.3, the aggregation function $f(\cdot)$ in Section 2.1 is removed, since it is inapplicable and unnecessary to the sequence-level reward function. For the minimalist LM training objectives Eqs. (5) and (6) in Section 2.2, we change them to the corresponding versions that use sequence-level guidance. Self-normalization in reward-weighted MLE Eq. (6) is removed, since it is again inapplicable and unnecessary to the sequence-level setting.

In this section, we continue our discussion in the Ablation Study (Section 4.3) by answering the following additional questions on our method.

**(a):** *Is our method robust to the hyperparameter(s): temperature $\beta$ and balancing coefficient $\alpha$?*

To study the choice of the temperature parameter $\beta$ in the soft-maximum/minimum aggregation Eq. (4), we vary the value of $\beta$ in the MIN variant in Tables 1 and 2 from $\beta = 2$. Furthermore, to study the balancing coefficient $\alpha$ in the REINFORCE-style LM-training approach Eq. (5), we vary the $\alpha$ parameter in the AVG variant in Table 1 from $\alpha = 2^{-3}$. Fig. 6 respectively shows the prompt results on the SST-2 dataset and the summarization results on the CNN/DM dataset. For summarization, we again plot the average ROUGE scores, with the breakdown scores of the three ROUGE metrics in Table 10 below.

Recall that the best baseline result on SST-2 in Table 1 is 90.5, and on CNN/DM in Table 2 is 31.3. We see that our method can achieve competitive results on a relatively wide range of the temperature $\beta$. A too-small value of $\beta$, such as 0.25 and 0.5, may incur a harder optimization problem and thus an inferior performance on both prompt and summarization tasks.

For the choice of the balancing coefficient $\alpha$, we see that our method provides competitive results in a relatively wide range of $\alpha \in [0.08, 0.15]$, when compared to the best baseline result of 90.5 in Table 1. A too-small value of $\alpha$ may not prevent the REINFORCE-style method from pre-mature convergence. The resulting LM therefore may not sufficiently explore the sampling space or capture multiple good behavior-modes, resulting in an inferior and highly varying performance. A too-large value of $\alpha$ distracts the optimization of the LM, and again leads to a worse result.

Table 10: Scores on each ROUGE metric for the summarization task on CNN/DM, where we vary the temperature parameter $\beta$ in the *soft-minimum* aggregation Eq. (4). The reported numbers are mean (standard deviation) over three random seeds. The row "Average" shows the average of the three ROUGE scores, *i.e.*, (ROUGE-1 + ROUGE-2 + ROUGE-L) / 3.

|  | $\beta = 2^{-2}$ | $\beta = 2^{-1}$ | $\beta = 2^0$ | $\beta = 2^1$ | $\beta = 2^2$ |
|---|---|---|---|---|---|
| ROUGE-1 | 40.77 (0.11) | 40.74 (0.09) | 40.79 (0.11) | 40.78 (0.06) | 40.80 (0.01) |
| ROUGE-2 | 18.67 (0.06) | 18.68 (0.05) | 18.68 (0.09) | 18.67 (0.03) | 18.71 (0.04) |
| ROUGE-L | 38.00 (0.10) | 37.98 (0.08) | 38.03 (0.12) | 38.01 (0.04) | 38.02 (0.01) |
| Average | 32.48 | 32.47 | 32.50 | 32.49 | 32.51 |

**(b):** *How does our method perform in generating longer prompts compared with the baseline?*

To further validate the harm of the delayed-feedback issue to the related LM-training methods that learn under the sparse sequence-level feedback, we compare our method with RLPrompt [60] on generating prompts with length increased from 5 to 10 and to 20 tokens, on the SST-2 dataset. Table 11 below shows the results.

Table 11: Test accuracy on the prompt task on the SST-2 dataset, for our method and RLPrompt on generating prompts with a length of 5, 10, and 20 tokens. We report the mean and standard deviation over three random seeds.

|  | RLPrompt | Ours (AVG) | Performance Gap |
|---|---|---|---|
| 5 Tokens | 90.5 (1.5) | 92.6 (1.7) | 2.1 |
| 10 Tokens | 75.8 (7.6) | 86.0 (2.9) | 10.2 |
| 20 Tokens | 65.2 (6.0) | 80.9 (4.5) | 15.7 |

We see that RLPrompt performs worse than our method on generating longer prompts. In particular, the performance gap increases as the prompt length (feedback delaying) increases. This comparison can further demonstrate the harm of the delayed-feedback issue in training text-generation LMs, and that our framework, in particular our preference-grounded token-level guidance for LM training, is a viable solution to it.

It is intrigued that the results of both methods deteriorate with the prompt length. After checking the generated prompts from our method, we find that longer prompts mostly contain many repeated tokens, as shown by the following example prompt of length 20

```
PerformanceExceptionMovieMovieMovieMovieMovieMovieMovieVideoVideoVideoVideo\
VideoVideoVideoImageVideoImageImage
```

which is separated into two lines at the location of "\" due to the page-width limit. In this prompt example, the tokens `Movie` and `Video` are each consecutively repeated seven times, and the bi-gram `ImageVideo` is repeated two times. Such prompts with heavy repetitions may confuse the downstream classifier.[6] This aligns with our intuition that a clear and succinct instruction is preferable than a long but verbose one.

As a side note, in generating Table 11, we use the default hyperparameters for both our method and RLPrompt. It is possible that RLPrompt requires careful tuning for generating longer prompts, due to the delayed-feedback issue that we try to address. We leave a thorough tuning of RLPrompt on long-prompt generation as a future work.

**(c):** *Is the efficacy of our framework tied to the specific preference sources considered in Section 4?*

To investigate whether the performance of our framework is tied to the specific preference-sources considered in the experiment section (Section 4), inspired by RL4LMs [61], we simulate the sequence-level preference on the summarization task by using another two automatic metrics "Rouge-avg" and "Rouge-avg2", rather than the classical Meteor score [90] in Section 4. Table 12 below presents the ROUGE scores of our method under each of the three preference sources on the CNN/DM dataset under the T5-base LM. For a more thorough investigation, we provide the results for our method both with and without the guidance re-estimation scheme. The baseline results in Table 12 below come from the best baseline method in Table 4 of Appendix A.1.

---

[6]A detailed description of the prompt task is deferred to Appendix D.

Table 12: Results for our method on CNN/DM summarization under T5-base LM when using different automatic metrics to simulate the sequence-level preference. We provide the detailed ROUGE scores for our method both with and without guidance re-estimation. "Baseline" denotes the results of the best baseline method in Table 4 of Appendix A.1. The reported numbers are the mean over three random seeds. The row "Average" shows the average of the three ROUGE scores, *i.e.*, (ROUGE-1 + ROUGE-2 + ROUGE-L) / 3.

| | Baseline | With Guidance Re-estimation | | | Without Guidance Re-estimation | | |
| --- | --- | --- | --- | --- | --- | --- | --- |
| | | Rouge-avg | Rouge-avg2 | Meteor | Rouge-avg | Rouge-avg2 | Meteor |
| ROUGE-1 | 42.9 | 43.14 | 43.07 | 43.09 | 42.96 | 42.98 | 42.98 |
| ROUGE-2 | 19.4 | 20.18 | 20.12 | 20.17 | 20.07 | 20.05 | 20.09 |
| ROUGE-L | 36.1 | 39.93 | 39.89 | 39.99 | 39.80 | 39.77 | 39.87 |
| Average | 32.8 | 34.42 | 34.36 | 34.42 | 34.28 | 34.27 | 34.31 |

Concretely, these two new automatic metrics "Rouge-avg" and "Rouge-avg2" are constructed as

$$\text{Rouge-avg} = 0.5 \times \text{ROUGE-1} + 0.5 \times \text{ROUGE-2} + 0.5 \times \text{ROUGE-L},$$
$$\text{Rouge-avg2} = 0.5 \times \text{ROUGE-1} + 0.5 \times 2 \times \text{ROUGE-2} + 0.5 \times \text{ROUGE-L},$$

where the "Rouge-avg" metric is exactly the same as that in the RL4LMs [61]. The "Rouge-avg2" metric is constructed by multiplying ROUGE-2 by 2 to make its numerical value similar to the others.

It is clear that changing the preference source from Meteor to these two alternative metrics does not significantly alter the performance of our method, especially when compared to the performance improvement of our method over the best baseline method in Table 4 of Appendix A.1. This set of comparisons confirms that the efficacy of our framework is generally not tied to a specific preference source. It could also further corroborate the effectiveness of our preference-grounding perspective on guiding the LM training.

# B  Additional Experiment Details

## B.1  Prompt Generation

**Implementation Details.**    To ensure a fair comparison, the implementation of our framework is based on the official codebase of RLPrompt available at https://github.com/mingkaid/rl-prompt, and the Hugging Face library [73]. We have provided some implementation details in Section 4.1. Here we continue the discussion.

The LM $\pi_\theta$ is parametrized as a frozen distilGPT-2 model with parameter $\theta$ being one MLP-layer of size 2048 inserted right before the output head. The token-level reward function $r_\phi$ is implemented as a distilGPT-2 with a two-layer projection-MLP of sizes 2048 and 1 on top. The LM $\pi_\theta$ is trained by a maximum of 12000 steps, with early stopping based on the validation set. The reward training is reconducted every 1000 steps during the first 6000 steps of the LM training process and is (and almost always) early stopped. RoBERTa-large is used [9] as the pre-trained downstream LM $\pi_{\text{DLM}}$.

**Datasets.**    We use the standard datasets provided in the RLPrompt codebase [60]. We test on three popular few-shot classification datasets in prior work [*e.g.*, 77, 78], *i.e.*, two sentiment binary-classification datasets SST-2 [79] and Yelp Polarity [81], and the topic four-way-classification dataset AG News [81]. In keeping with the standard few-shot setting [76], both the training and the validation sets have 16 examples per class. To mitigate the randomness in the few-shot setting, each dataset is subsampled into five few-shot training and validation sets, while the test set is standard. We train our models on each few-shot (sub-)dataset with three random seeds and evaluate three generated prompts in each case. For all three tested datasets, we report the average test accuracy and standard deviation across all evaluated prompts in all random seeds and all few-shot (sub-)datasets.

**Hyperparameters.**    Apart from the hyperparameters discussed in the ablation study (Section 4.3 and Appendix A.2), most other hyperparameters as well as the training and evaluation procedures of our framework follow RLPrompt. Additionally, we list the important hyperparameters for training our reward model in Table 13, and important hyperparameters for training our LM in Table 14. The generated prompts have a fixed length of 5. The same hyperparameters are used in all tested datasets.

**Baselines.**    For the baseline results in Table 1, we rerun the codebase of RLPrompt under the same random seeds and evaluation script as our method. Other baseline results are from the literature

[60, 88]. We note that our reported RLPrompt results have some small discrepancies compared to the original paper's results. We have confirmed our reproduced results with RLPrompt's authors and with Table 2 of the recent TEMPERA paper [88].

Table 13: Hyperparameters for training our reward model in the prompt-generation task.

| Hyperparameter | Value |
| --- | --- |
| Gradient clipping norm | 5.0 |
| Max train steps | 10000 |
| Steps per epoch | 100 |
| Number of epochs | 100 |
| Learning rate | 5e-5 |
| Batch size | 64 |
| Learning-rate decay | 0.8 |
| Learning-rate scheduler | `ReduceLROnPlateau` |
| Scheduler patience | 2 |
| Early-stop count | 7 |
| Optimizer | Adam [95] |
| Backbone | distilGPT-2 |

Table 14: Hyperparameters for training our LM in the prompt-generation task.

| Hyperparameter | Value |
| --- | --- |
| Gradient clipping norm | 5.0 |
| Max train steps | 12000 |
| Steps per epoch | 500 |
| Number of epochs | 24 |
| Learning rate | 5e-5 |
| Batch size | 32 |
| Learning-rate decay | 0.8 |
| Learning-rate scheduler | `ReduceLROnPlateau` |
| Scheduler patience | 2 |
| Early-stop count | 7 |
| Optimizer | Adam |
| Backbone | distilGPT-2 |
| Reward retrain period | 1000 steps |

## B.2  Text Summarization

**Implementation Details and Hyperparameters.**  The implementation of our framework is based on the Hugging Face library [73]. We have provided some implementation details in Section 4.2. The discussion is continued here.

Due to our limited computational resources, unless explicitly mentioned, we use the standard T5-small model [89] for the LM. Similar to the prompt tasks, the token-level reward function is implemented also as a T5-small model, with a two-layer projection-MLP on top with sizes 2048 and 1. The LM $\pi_\theta$ is trained for a standard 5 epochs. Apart from the hyperparameters discussed in the ablation study (Section 4.3 and Appendix A.2), most other hyperparameters as well as the training and evaluation procedure of our framework follow the standard setting of using a T5 model for text summarization on the Hugging Face library. Additionally, we list the important hyperparameters for training our reward model in Table 15, and important hyperparameters for training our LM in Table 16. The same hyperparameters are used in both the CNN/DailyMail and the XSum datasets.

We note that the ROUGE-L metric we report is technically the `rougeLsum` metric from the Hugging Face interface and in the RL4LMs' codebase [61]. This one matches the result scales in prior work especially on texts with newlines ("\n"), as reported in this GitHub issue.

**Baselines.**  For the baseline methods' results in Table 2, we rerun the codebase of RL4LMs [61] with a T5-small model as our method. We have carefully tuned the (supervised+) PPO/NLPO in RL4LMs on several hyperparameters, such as `learning_rate`, `kl_div:coeff`, `kl_div:target_kl`, and so on. Furthermore, we ran these baseline methods on the same random seeds as our method and we provide error bars. Since we use the T5-small model and the same random seeds for both our method and the baselines, our reported results are therefore (more) fair comparisons.

Table 15: Hyperparameters for training our reward model in the text-summarization task.

| Hyperparameter | Value |
|---|---|
| Gradient clipping norm | 5.0 |
| Number of epochs | 1 |
| Amount of training data | 10% of training set |
| Learning rate | 5e-5 |
| Batch size | 32 |
| Optimizer | Adam |
| Backbone | T5-small |

Table 16: Hyperparameters for training our LM in the text-summarization task.

| Hyperparameter | Value |
|---|---|
| Gradient clipping norm | 5.0 |
| Number of epochs | 5 |
| Learning rate | 5e-5 |
| Batch size | 32 |
| Optimizer | AdamW [96] |
| Weight decay | 0.0 |
| Backbone | T5-small |
| Reward retrain period | 0.5 epoch |

## C  A Naïve Numeric Example for the *Average* Aggregation

This section provides a naïve numeric comparison that the *average* aggregation in Section 2.1 will not automatically favor longer sequences, while the classical *summation* will.

Suppose we have $K = 2$ sequences $\tau^1$ and $\tau^2$ for preference learning, respectively having length $T^1 = 5$ and $T^2 = 15$. For simplicity, assume that all tokens in $\tau^1$ and $\tau^2$ are the same and all have reward 1, *i.e.*, $r_\phi(s_t^k, a_t^k) = 1, \forall k, t$. The average sequence length $C$ is then $C = (1/2) \times (5 + 15) = 10$. For the first sequence $\tau^1$, the *average*-aggregated sequence-level evaluation $e_\phi^{\mathrm{avg}}(\tau^1) = (10/5) \times \sum_{t=0}^{4} 1 = (10/5) \times 5 = 10$. And for the second sequence $\tau^2$, $e_\phi^{\mathrm{avg}}(\tau^2) = (10/15) \times \sum_{t=0}^{14} 1 = (10/15) \times 15 = 10$. Therefore, no sequence will be automatically preferred based only on the length.

By contrast, when using the classical *summation* as the aggregation function, $\tau^1$ will be evaluated as $\sum_{t=0}^{4} 1 = 5$ while $\tau^2$ will be evaluated as $\sum_{t=0}^{14} 1 = 15$. So, indeed, the longer sequence $\tau^2$ will be automatically preferred.

## D  Details on the Prompt Generation Task

**Task Description.**  In discrete text-prompt generation [*e.g.*, 10, 74], we input a discrete text-prompt $a$ and an observation sequence $o$ to a large pre-trained downstream LM $\pi_{\mathrm{DLM}}(y_{\mathrm{DLM}} \mid a, o)$ to directly classify text $o$, without finetuning $\pi_{\mathrm{DLM}}$. Here, $y_{\mathrm{DLM}}$ denotes the output of the large downstream LM $\pi_{\mathrm{DLM}}$ on the observation text $o$ prompted by text $a$. We follow the classical prompt setting [*e.g.*, 10, 75, 60] that solves the classification problem by an encoder-only downstream LM via token infilling. Classification is reduced to selecting tokens corresponding to some predefined class labels, known as verbalizers, such as "happy" for positive and "sad" for negative. The set of verbalizers is denoted as $\mathcal{C}$. As an example, to classify an observation text $o$ by prompt $a$ using an encoder-only downstream LM $\pi_{\mathrm{DLM}}$, we input a template such as "[o] [a] [MASK]" to $\pi_{\mathrm{DLM}}$, and select the most probable verbalizer token that fills into [MASK].

**Setting.**  In our input-agnostic setting, the generated prompt is independent of the observation text $o$. During inference time, only the learned prompts are used and the LM $\pi_\theta$ is discarded. The initial input $x$ to $\pi_\theta$ is a dummy, and the target $y$ is the class label in the mask position. We also adopt the few-shot setting, where the training set consists of a small number of samples per class. There is a larger standard test set for evaluation. With a fixed length $T$, the goal is to find discrete text-prompts $a = (a_0, \ldots, a_{T-1})$ that have high test accuracy.

**Source of the Preference.**  For learning the token-level guidance, we simulate the sequence-level preference by the recently proposed stepwise metric $\mathcal{R}_{\mathrm{step}}$ in Deng et al. [60], *i.e.*, the higher the metric value the better prompt. This choice ensures a fair comparison with RLPrompt [60] and avoids a potential overfitting that we train and evaluate the LM on the same evaluation metric "accuracy".

Given a prompt $a$, observation text $o$, and the true class label $y \in \mathcal{C}$, $\mathcal{R}_{\mathrm{step}}$ measures the gap between the true class's probability and the highest probability in other classes. The gap is defined as

$$\mathrm{Gap}_o(a, y) = \pi_{\mathrm{DLM}}(y \mid a, o) - \max_{y' \in \mathcal{C}, y' \neq y} \pi_{\mathrm{DLM}}(y' \mid a, o),$$

where $\text{Gap}_o(\boldsymbol{a}, y) > 0$ when the prediction $y_{\text{DLM}}(\boldsymbol{a}, o)$ for text $o$ is correct and $< 0$ otherwise. Define the indicator for correct prediction for $o$, $\text{Corr}_o$, as $\text{Corr}_o = \mathbf{1}\{\text{Gap}_o(\boldsymbol{a}, y) > 0\}$. The stepwise metric $\mathcal{R}_{\text{step}}$ for prompt $\boldsymbol{a}$ on observation text $o$ and true class label $y$ is define as

$$\mathcal{R}_{\text{step}}(y_{\text{DLM}}(\boldsymbol{a}, o), y) = \lambda_1^{1-\text{Corr}_o} \lambda_2^{\text{Corr}_o} \times \text{Gap}_o(\boldsymbol{a}, y),$$

where $\lambda_1 = 180$ and $\lambda_2 = 200$. In the experiments (Section 4 and Appendix A.2), we report test accuracy as in prior works.

**LM Training.** Since the prompt-generation task does not assume the availability of supervised data — the ground-truth prompts, the LM $\pi_\theta$ is trained by the REINFORCE-style update in Section 2.2 to automatically discover highly-accurate prompts.

# E More Related Work

**Prompt Generation.** Prior works [*e.g.*, 6, 10, 85, 97] have shown that manual prompts can steer LMs to perform NLP tasks in the few/zero-shot setting. In general, prompts can be discrete, consisting of real token-strings; or can be continuous, where the prompts are entirely free word-embeddings that do not map to real tokens. Several works [*e.g.*, 98–102, 83] tune continuous soft prompts using gradient descent, which typically requires some expensive gradient information [78, 103]. In this work, we apply our framework to the task of input-agnostic discrete-prompt optimization due to its challenging setting, better human understandability of the learned prompts [104, 105], potential transferability across LMs [106, 76, 60], and more robustness in the low-data regime [99]. Recent works propose some new settings such as input-dependent prompt-tuning [88], which are potential further applications of our framework and are left for future work.

**Text Summarization.** Apart from using RL techniques discussed in Sections 3, prior works on text summarization [*e.g.*, 7, 89, 107–109] mainly focus on structural designs of the LMs and improvements on the source of the (pre-)training data, where the LMs are typically trained by vanilla MLE on the supervised data. In this paper, we apply our preferenced-grounded token-level guidance to this task by considering a weighted-MLE objective for LM training. The weights given by the learned reward function reflect some sequence-level preference among multiple candidate summaries. Our framework thus has the potential to learn and improve from lower-quality data, and generate summaries fulfilling more general evaluation metrics, such as human preference.

**Weighted MLE in NLP.** Though not a very common techinique, the approach of weighted MLE has been adopted in prior NLP research. For example, RAML [110] samples outputs proportionally to its exponentiated scaled "reward" (negative edit/Hamming distance) using stratified sampling. GOLD [111] frames text generation as an offline RL problem with expert demos and learns from the demos by importance weighting, where training examples with higher probability under the model are weighted higher. Besides, Ghosh et al. [112] apply the weighted MLE technique to table-to-text generation and Junczys-Dowmunt et al. [113] apply this technique to grammatical error correction for machine translation. Our token-level reward-weighted MLE in Section 2.2 adds to this research thread by emphasizing the important tokens in the supervised sequences and downweighting the unimportant tokens. This design may better utilize the LM capacity and the optimization budget. The efficacy of our reward-weighted MLE is experimentally verified in Section 4.2.

**Align LMs with Preference.** Similar to our paper, prior works on aligning LMs with preference typically focus on adjusting the pretrained LMs, where preference comes from human feedback or from some automatic metrics. A classical strategy is to add external filters on top of the pretrained LMs for the generated text sequences or for the training sequences [*e.g.*, 19], where the LMs are trained using MLE on abundant supervised data. Another classical approach finetunes LMs using supervised learning (vanilla MLE) on some curated/improved datasets [20–22], or on massive highly-curated collections of tasks phrased as instructions for supervised finetuning the LMs [114–117]. Apart from supervised learning, reinforcement learning techniques have also been applied to learn from human feedback (RLHF). Similar to the discussion in Section 3, these works typically learn a *sequence-level* classifier that predicts human (pairwise) preferences and during LM training add a general-purpose KL penalty that is less-targeted to the specific LM task and feedback (preference, metric scores, *etc.*) [*e.g.*, 14, 23–25], such as a token-level KL penalty towards the initial LM prior to training.

Alternatively, the divergence of the LMs from a target distribution can also be used as the finetuning objectives. This line of research [*e.g.*, 118–120] formalizes controlled text generation as a constraint satisfaction problem over LM's probability distribution, with an additional divergence-minimization objective that the LMs should have a minimal KL- or $f$-divergence from the original pretrained model. These approaches, however, require explicit functional specification on the constraints or on the human preference, rather a more vague form of (binary) comparison between LM samples. For example, Go et al. [120] consider human preference as a probability distribution measuring how well the generated text-sequence satisfies the preference. Apart from this more demanding requirement, these approaches further require special methods to sample from the resulting LM.

To sum up, prior works on aligning LMs with preference mostly focus on an ungrounded *sequence-level* guidance, which can suffer from the delay-feedback issue in LM training, as discussed in Sections 1 and 3. By contrast, our preference-grounding perspective can provide a stable, data-driven, task-specific *token-level* guidance on LM training, and can potentially improve on the vanilla MLE, especially when the quality of the supervised data cannot be guaranteed. We experimentally validate this intuition in Section 4 and Appendix A.2.

Apart from fine-tuning the pretrained LMs, Korbak et al. [121] recently apply preference alignment to the pre-training stage of the LMs. As with prior works, the sparse sequence-level evaluation (without KL penalty/stabilizer) is directly used, to learn a token-level value function, to condition the LM generation on, or for a reward-weighted regression objective. The pre-training stage in Korbak et al. [121] is a potential further application of our framework since we make no assumption on the zero-shot ability of the initialized LMs, as discussed in Sections 2.2 and 4.3.

We also notice that a recent robotics paper [122] proposes to *learn* a *weighted-sum* aggregation together with the per-step reward, to form the sequence-level evaluation in learning the reward function, based on pairwise preference over two trajectories of equal length. Compared with this recent work, our aggregation functions in Section 2.1 do not require additional modeling and training, and therefore can be more efficient and more stable for the reward-function learning. Additionally, we do not assume that trajectory lengths are equal, as this may be infeasible for LM tasks such as text summarization. Furthermore, our framework allows utilizing the preference among more than two trajectories, rather than the classical pairwise preference. In this particular aspect, our framework can be more general than this recent work of Kim et al. [122].

# F  A Discussion on Applying RL Methods to LM Tasks

## F.1  LM Generation as a Token-level MDP

In most LM generation tasks, there is a dataset $\mathcal{D} = \{(x^i, y^i)\}_{i=1}^N$ of $N$ supervised examples, where $x$ is the input to the LM that can be a dummy, and $y \in \mathcal{Y}$ is the target text sequence. Viewing the LM as a token-level RL policy, LM generation can be formulated as a sequential decision-making problem, specified by the Markov Decision Process (MDP) $\mathcal{M} = (\mathbb{S}, \mathbb{A}, P, \mathcal{R}, \gamma, \mu_0)$ [123]. Specifically, $\mathbb{S}$ is the state space, where the state at timestep $t$, $s_t$, consists of the LM input $x$ and the previously generated tokens $a_{<t} = (a_0, \ldots, a_{t-1}), t > 0$, *i.e.*, $s_0 = x$ and $\forall t > 0, s_t = (x, a_{<t})$. $\mathbb{A}$ is the action space, which is the vocabulary $\mathcal{V}$, and an action $a_t$ at timestep $t \geq 0$ is a token from $\mathcal{V}$. $P(s_t, a_t) : \mathbb{S} \times \mathbb{A} \to \mathbb{S}$ is the transition function that deterministically appends the newly sampled token to the end of the current state, *i.e.*, $\forall t \geq 0, s_{t+1} = (s_t, a_t) = (x, a_{\leq t})$. $\mathcal{R}(s_T, y) : \mathbb{S} \times \mathcal{Y} \to \mathbb{R}$ is the environmental reward (task-specific evaluation metric) that depends on the *final state* $s_T$ of the LM-generation trajectory and the target sequence $y$. Here $T$ is the ending time of the trajectory, *i.e.*, the length of the full generated text sequence; and $s_T = (x, a_0, \ldots, a_{T-1})$ is the final state of the generation trajectory consisting of the LM input $x$ and the full generated text sequence $\boldsymbol{a} = (a_0, \ldots, a_{T-1})$. $\gamma \in [0, 1]$ is the discount factor. And $\mu_0(x) : \mathbb{S} \to [0, 1]$ is the distribution of the initial input $x$.

We denote the LM as $\pi_\theta(a_t \mid s_t)$, parametrized by $\theta$. At each timestep $t$, $\pi_\theta(a_t \mid s_t)$ generates the next token $a_t$ given the current state $s_t = (x, a_{<t})$. The ultimate goal of policy learning (LM training) is to maximize the expected environmental reward $\mathcal{R}$, which can be expressed as

$$\max_\theta \mathbb{E}_{(x,y)} \mathbb{E}_{\boldsymbol{a} \sim \prod_{t=0}^{T-1} \pi_\theta(a_t \mid s_t)} [\mathcal{R}(s_T = (x, \boldsymbol{a}), y)] \,,$$

where $(x, y)$ is drawn from the corresponding sampling distribution.

## F.2 Delayed Feedback in RL-based LM Training

As discussed in Appendix F.1, the environmental reward $\mathcal{R}(s_T, y)$ is only defined on the full generated text sequence $a$. The token-level MDP formulation of LM generation thus meets the problem of sparse reward-signal or the delayed feedback issue discussed in Section 1. Hereafter, we will use "sparse reward (signal)" and "delayed feedback" interchangeably depending on the context, as they are used synonymously in the RL literature.

Specifically, prior works [e.g., 31, 60, 32] often manually interpolate the intermediate rewards by some non-informative values such as 0 or $-1$, i.e., $\forall\, t \geq 0$

$$\mathcal{R}(s_t, y) = \begin{cases} 0 \text{ or } -1, & t < T \\ \mathcal{R}(s_T, y), & t = T \end{cases}. \tag{7}$$

It is clear that the reward signal is sparse. In other words, the feedback to intermediate actions/tokens is delayed until the full text-sequence has been generated.

We note that this sparse-reward/delayed-feedback problem will not be addressed by the standard actor-critic or Q-learning methods in RL. With only sparse reward-signals, it can be difficult to estimate the token-level value functions in these RL methods.

Specifically, the standard Monte Carlo estimate of the value functions is known to have high variance due to the large sampling space [123]. This problem is even severe in the LM tasks where there are exponentially many text sequences that can follow a partial sequence.

Further, as discussed in Guo et al. [31], the sparse-reward/delayed-feedback problem can also hurt the bootstrapping-style method for learning the value functions, since the standard value-function learning can suffer from "the unstable per-step bootstrapping-style training with sparse reward signals." This can subsequently harm the LM training since many actor-critic or Q-learning methods rely heavily on how accurately the learned value function(s) can assess the quality of intermediate text sequences [31, 123, 124].

## F.3 Sparse Reward with KL Penalty

With the sparse-reward/delayed-feedback issue in Appendix F.2, prior works typically add a token-level KL-penalty to the sparse sequence-level environmental rewards Eq. (7). For simplicity, assume that in Eq. (7) the intermediate rewards are interpolated by 0. The KL-stabilized reward signal $R(s_t, a_t, y)$ is

$$R(s_t, a_t, y) = \begin{cases} -c \cdot \text{KL}(\pi_\theta(a_t \,|\, s_t) \,||\, \pi_0(a_t \,|\, s_t)), & t < T - 1 \\ \mathcal{R}(s_T, y) - c \cdot \text{KL}(\pi_\theta(a_t \,|\, s_t) \,||\, \pi_0(a_t \,|\, s_t)), & t = T - 1 \end{cases}, \tag{8}$$

where $c$ is a hyper-parameter and $\pi_0$ is some prior distribution, such as the uniform distribution [31, 60], the initial LM prior to training [23, 61], the supervised-fine-tuned model [62, 63, 12, 14], or the base momentum model [65]. For a concrete example, see Line 224-235 of the popular trlx package's implementation.

With this KL-stabilized reward signal $R(s_t, a_t, y)$, the action-value function for the policy/LM $\pi_\theta$ is

$$
\begin{aligned}
Q(s_t, a_t, y) &= \mathbb{E}_{\{a_{t'}\}_{t'=t+1}^{T-1} \sim \pi_\theta} \left[ \sum_{t'=t}^{T-1} \gamma^{t'-t} R(s_{t'}, a_{t'}, y) \,|\, s_t, a_t \right] \\
&= \mathbb{E}_{\{a_{t'}\}_{t'=t+1}^{T-1} \sim \pi_\theta} \left[ \gamma^{T-1-t} \mathcal{R}(s_T, y) - c \cdot \sum_{t'=t}^{T-1} \gamma^{t'-t} \text{KL}(\pi_\theta(a_{t'} \,|\, s_{t'}) \,||\, \pi_0(a_{t'} \,|\, s_{t'})) \,|\, s_t, a_t \right]
\end{aligned}
\tag{9}
$$

It is clear from Eq. (9) that the environmental reward $\mathcal{R}(s_T, y)$ is multiplied by a factor exponentially decayed with respect to the length of the remaining horizon $T - 1 - t$. Without the KL penalty, the action-value $Q(s_t, a_t, y)$ could be tiny when $t$ is small, i.e., at the beginning of the text-sequence generation. This could make it hard to accurately model and learn the action values, echoing the previously-stated harm of the sparse-reward/delayed-feedback problem mentioned by Guo et al. [31]

Recall that the standard actor-critic and Q-learning methods in RL use the action-value function $Q(s_t, a_t, y)$ as the token-level guidance (per-step critic) for policy/LM training. Due to the exponentially decaying factor $\gamma^{T-1-t}$, when the discount factor $\gamma$ in Eq. (9) is not sufficiently large, this

token-level guidance $Q(s_t, a_t, y)$ in RL-based LM training mainly reflects the (discounted) sum of future KL-penalty, rather than the actual goal of LM training — the environmental reward $\mathcal{R}(s_T, y)$. This phenomenon can be more evident at the beginning of the text-sequence generation, *i.e.*, when the length of the remaining horizon $T - 1 - t$ is long. On the other hand, learning the action-value function $Q(s_t, a_t, y)$ under a large discount factor $\gamma$ is known to be challenging [123], since the highly varying (late) future can significantly affect the current action value $Q(s_t, a_t, y)$. The selection of the discount factor $\gamma$, therefore, becomes a tradeoff and a challenge. Note that $\mathcal{R}(s_T, y)$ here is generic and can represent automatic evaluation metrics or (human) preference, and that the beginning of text generation can affect all subsequent token selections. Intuitively, using Eq. (9) as the token-level guidance for policy/LM training can thus be less successful in the concrete LM task, especially when generating longer sequences, as we verified in Appendix A.2.

In the experiments (Section 4 and Appendix A.2), we compare our preference-grounding approach with RL-based baselines that estimate a standard value function similar to Eq. (9) from sparse environmental reward with KL penalty, such as the RLPrompt method [60] and the (supervised+) PPO/NLPO methods in RL4LMs [61]. We leave as future work the potential combination of our preference-grounded guidance with actor-critic and Q-learning methods in RL-based LM training.

## G   Further Discussion on the Guidance Re-estimation Scheme

As discussed in Section 2.2, in this paper, we deal with the most general setting where the LM training directly starts from a raw pre-trained LM, rather than an initial LM that has been fine-tuned via supervised learning on the desired dataset, such as in Stiennon et al. [12]. We also make no assumptions about the zero-shot ability of the raw pre-trained LM. We choose this setting because it is more general and naturally fits into the task of text-prompt generation, where supervised datasets of good prompts are not available and the initial LM cannot generate good prompts.

As discussed before, under this general setting, the LM $\pi_\theta$ can evolve from a less-preferred distribution to a highly-preferred one, over the training process. Since our reward function $r_\phi$ is trained by text sequences sampled from $\pi_\theta$, there is a distribution shift between the sequences used to train $r_\phi$ during reward-function learning, and the sequences evaluated by $r_\phi$ during LM training, especially after $\pi_\theta$ has been sufficiently improved. To keep $r_\phi$ as accurate guidance for LM training, a natural idea is to refine $r_\phi$ periodically on the text generations from the latest LM, leading to our reward-function retraining scheme.

We emphasize that *the reward-function retraining scheme does not give our method an unfair advantage over the baseline methods*. In particular, RLPrompt [60] and RL4LMs' methods [61] retrain their value-functions in every optimization step, and thus, they query the environmental reward in every optimization step. Specifically, in Algorithm 1 of the RL4LMs paper, the penalized reward $\hat{R}_t$ is calculated in each optimization step, whose calculation requires the true environmental reward $R$ (Eq. (1) of the RL4LMs paper). Besides, in the codebase of RLPrompt, this environmental interaction is implemented in this line, which is queried in every optimization step, as seen in this line. In the notion of Reinforcement Learning from Human Feedback (RLHF), this every-step interaction is similar to asking humans to score the LM generations in every training step, which can be infeasible. By contrast, in our paper, we reduce the frequency of these environmental interactions by retraining the guidance model only periodically and only during the first half of the LM-training process.

Though the motivation of this reward-function retraining scheme comes from model-based RL (Section 2.2), we notice that some prior RLHF works do implement similar ideas. For example, Page 2 of Ziegler et al. [23] mentions that "..., we continue to collect additional data and retrain our reward model as the policy improves (online data collection)." Page 2 of Stiennon et al. [12] mentions that "We can then gather more human data using samples from the resulting policy, and repeat the process." Page 5 of Menick et al. [25] and Page 20 of Bai et al. [24] also have similar discussions. Based on these, our reward-function retraining scheme is both well-motivated and practical, even with human rankings in RLHF.

## H   Potential Negative Societal Impacts

Since our framework can ground the sequence-level preference into token-level guidance for LM training and can be not tied to a specific preference source, it is possible that this framework may be

used to train ill-intended LMs by grounding some malicious or unethical preferences. This potential negative impact may be mitigated by closer monitoring the datasets on which our framework operates.

# I   Limitations

Since our token-level guidance is learned by grounding sequence-level preference, a potential failure case of our framework will be when the preference orderings are very noisy. In this situation, the learned guidance may not be meaningful and hence could even deteriorate the subsequent utilization of it in LM training.

Even though we have shown in Section 4.3 that it can be beneficial to use more than two sequences to learn the token-level guidance, it can be practically challenging to obtain a high-quality ranking among many candidate text sequences, *e.g.*, when the number of sequences is more than seven.

Besides, the reward-function retraining scheme may incur some additional computational complexity, compared with training the reward function only once and fixing it throughout the LM-training process.

# J   Computational Resources

The experiments are conducted on NVIDIA GeForce RTX 3090 and NVIDIA A100 GPUs. Depending on the specific task and setting, several models could be trained concurrently on a single GPU.

