# Appendix for "Preference-grounded Token-level Guidance for Language Model Fine-tuning"

## Table of Contents

# A   Additional Experimental Results

## A.1   Tabular Results

Table 3: Examples of the generated discrete input-agnostic text-prompt and their classification accuracy on the corresponding test set.

| SST-2 | | AG News | |
|---|---|---|---|
| Prompt | Accuracy | Prompt | Accuracy |
| guys filmmaker filmmaker rated Grade | 94.18 | newsIntroduction Comments Tags Search | 85.78 |
| MovieMovieFilm rated Grade | 94.18 | newsTopic Blog Support Category | 85.55 |
| Rated CinemaScoreReporting Grade | 94.01 | news RecentRecentPhotosIntroduction | 84.53 |
| employment theater rated Oscars Grade | 93.96 | news Recent Brief LatestExample | 84.51 |
| scene filmmaking rated comedian Grade | 93.85 | newsVirtualBlogBlogNet | 84.33 |

Table 4: Detailed results on CNN/DM summarization under T5-base LM for Section 4.2. We bold the best result of each metric. Baseline results are directly cited from RL4LMs [58]. "Env. Reward" denotes the environmental reward in RL4LMs. The "ROUGE-L" here refers to "Rouge-LSum" in RL4LMs and in the Hugging Face interface, which is discussed in details in Appendix B.2. In Section 4.2, we plot the results of our method with the *average* aggregation, which is the best variant in Table 2. We report the mean and standard deviation of our method over three random seeds.

| Algorithm | Env. Reward | ROUGE-1 | ROUGE-2 | ROUGE-L | Meteor |
|---|---|---|---|---|---|
| Lead-3 | | 40.1 | 17.5 | 36.3 | 33.3 |
| Supervised | | 41.1 | 17.7 | 34.3 | 30.9 |
| | Rouge-1 | 41.0 | 18.2 | 34.9 | 27.6 |
| PPO | Rouge-Avg | 39.6 | 17.6 | 33.8 | 27.0 |
| | Meteor | 40.8 | 17.8 | 34.2 | 30.1 |
| | Rouge-1 | 40.4 | 18.0 | 34.4 | 27.5 |
| NLPO | Rouge-Avg | 40.4 | 17.7 | 34.4 | 27.4 |
| | Meteor | 40.5 | 18.0 | 34.3 | 29.2 |
| | Rouge-1 | 41.7 | 18.9 | 35.8 | 27.8 |
| Supervised + PPO | Rouge-Avg | 42.5 | 19.4 | 36.3 | 29.6 |
| | Meteor | 42.6 | 19.4 | 36.1 | 31.6 |
| | Rouge-1 | 42.1 | 19.3 | 36.1 | 28.7 |