# OpenReview forum: "Preference-grounded Token-level Guidance for Language Model Fine-tuning"
_NeurIPS.cc/2023/Conference — NeurIPS 2023 poster_

### Official Review · Reviewer_9EYo · 2023-06-19

**Soundness:** 3 good
**Presentation:** 2 fair
**Contribution:** 3 good
**Rating:** 6
**Confidence:** 3

**Summary:**

This paper aims to tackle the misalignment between sequence-level preferences and token-level language model in NLG. The authors design an iterative training framework that integrates the sequence-level preference into token-level training guidance, mitigating the granularity mismatch. Experiments are conducted on two different LM tasks - discrete-prompt generation and text summarization, indicating its effectiveness.

**Strengths:**

1. The methodology of decomposing sentence-level preference into token-level preference presented in the paper is both intuitive and rational, providing a feasible approach to address the granularity mismatch in language model training.

2. Through comparative performance with baseline models, comprehensive ablation studies and discussions, the paper demonstrates the proficiency and potential benefits of the proposed model.

**Weaknesses:**

1. The paper lacks a precise definition of 'preference,' appearing to suggest it is human bias towards superior text quality. However, the method's experimental use of the METEOR score as a sequence-level reward contradicts this assumed understanding. This ambiguity may lead to confusion among readers and raises questions about the evaluation methodology. Why using METEOR as rewards can improve the evaluation in terms of ROUGE scores? Human evaluations should be conducted for a more comprehensive assessment.

2. The reported performance on CNNDM and XSum tasks, measured in ROUGE scores, is below benchmark levels. A basic BART model reportedly outperforms the proposed method (44 vs 40 ROUGE-1 score), which questions the effectiveness of the new training process. This raises concerns that the method may not improve upon existing architectures as suggested.

3. The absence of an accompanying code with the paper is a significant drawback. Without this, it is challenging to reproduce the results, especially given the complexity of applying reinforcement learning. This could limit the wider verification and applicability of the presented method.

**Questions:**

Can the performance be enhanced by integrating the token-level feedback with RL-based baseline models? These baseline models appear to show promising performance as depicted in Figure 2.

**Limitations:**

yes

---

> ### Author Rebuttal · Authors · 2023-08-06
>
> We deeply appreciate the reviewer for your thoughtful review. We would like to draw your attention to our **General Response** on human evaluation results on CNN/DM summarization and a discussion on why our summarization results are below the SOTA.
> Below, we address your remaining concerns in detail.
>
> > **Q1.** About the definition of “preference” in our paper.
>
> **A.** As discussed in Line 101-103, in this paper we make no assumption about the source of the preference --- it may come from human ranking or task-specific evaluation metrics. Our notion of preference is essentially an ordering of the text-sequences based on the evaluations of full text-sequences, where the evaluations come from automatic evaluation metrics or humans.
>
> Therefore, we believe that using the Meteor score to obtain the sequence-level preference in our experiments aligns with our notion of preference.
>
> We apologize for the confusion and will make this more clear in the next version of our draft.
>
> > **Q2.** Why using METEOR as rewards can improve the evaluation in terms of ROUGE scores?
>
> **A.** We are a bit confused about this question and will appreciate it if any further explanation can be provided. We provide our current best answer below.
>
> Meteor measures the matching between the model-generated string and the reference string, so we believe that it is a valid preference source for the summarization task.
>
> As shown in Section 4.2 and Table B in the added PDF, our method improves over the baselines in terms of several metrics: the ROUGE scores, Meteor score, and BertScore. As discussed in Section 4.2, we attribute our performance gain to our main algorithmic contribution: learning and utilizing the preference-grounded token-level guidance for LM training.
>
> Besides, we note that the baseline results from RL4LMs [1] suggest that using Meteor as the environmental reward for training the RL methods, (Supervised+) PPO/NLPO, can lead to ROUGE scores competitive or even stronger than using ROUGE-based environmental rewards.
> This further validates our use of Meteor as the preference source in the summarization task.
>
> [1] Ramamurthy, Rajkumar, et al. "Is Reinforcement Learning (Not) for Natural Language Processing?: Benchmarks, Baselines, and Building Blocks for Natural Language Policy Optimization." arXiv preprint arXiv:2210.01241 (2022).
>
>
> > **Q3.** There is no accompanying code.
>
> **A.** As stated in L​​ine 214, to facilitate the reviewing process, our source code has been anonymously released. The clickable link in the submitted paper is under the red word “released”. Since links are not allowed in the rebuttal text, we respectfully refer the reviewer to our paper for the link.
>
> We apologize for the confusion and will make the link more apparent in the next version of our draft.
>
>
> > **Q4.** Can integrating our token-level feedback with RL-based methods enhance the performance?
>
> **A.** Thank you for the great suggestion!
>
> Since our LM training objectives in Section 2.2 are only minimalist, we believe that more sophisticated training approaches, such as RL-based methods, can further improve our performance.
> On the other hand, as discussed in Appendix F.2, due to the granularity mismatch between the native sequence-level feedback and token-level LM training/generation, RL-based LM training can suffer from the delayed feedback issue. As a potential mitigation of this issue, it is promising to integrate our preference-grounded token-level guidance into RL-based methods for LM training.
>
> Nevertheless, we kindly note that since our paper is not aimed at RL-based LM training, due to the page limit, the combination between our token-level feedback and RL-based methods is out of this paper’s scope. As discussed in Line 341, we will certainly pursue this direction in our future work.

---

> > ### Comment · Reviewer_9EYo · 2023-08-16
> >
> > Thanks for your response. I've read the rebuttal and I'll keep my score.

---

> > > ### Author Response · Authors · 2023-08-16
> > > **Response to Reviewer 9EYo**
> > >
> > > Thank you so much for your time in reading our rebuttal! We wish our response could be helpful.

---

### Official Review · Reviewer_gmdH · 2023-07-05

**Soundness:** 3 good
**Presentation:** 3 good
**Contribution:** 3 good
**Rating:** 7
**Confidence:** 4

**Summary:**

To fine-tune an LM, the paper proposes “token-level guidance” by leveraging sequence-level preference. The algorithm alternates between two stages: (1) learning token-level “guidance” (aka reward function) and (2) fine-tuning LM using the “guidance”/reward.

To aggregate token-level rewards, the authors propose aggregation functions that are different from the classical summation: average, soft maximum, and soft minimum.

Figure 1 is a good illustration of the learning algorithm. It’s similar to RLHF, except the preference is based on multiple generated sequences, the reward is token-level, and the aggregation function can be different.

Two “minimalist” LM training objectives are proposed:
When there’s no supervised data, the authors use a REINFORCE-style update (with a max-entropy regularizer).
When we have enough data, the authors use a reward-weighted MLE objective – the weight depends the importance of the token, which is essentially the normalized token-level reward.

Experiments are done on prompt generation (generating discrete prompts so that the accuracy of the corresponding task is good) and text summarization. The performance is competitive with respect to baselines shown in Table 1 and Table 2.





**Strengths:**


It’s great that the authors have done ablation that removes reward retraining, as shown in Section 4.3 (b). Good to see that the performance, after removing retraining, is still competitive with respect to baselines.

The token-level reward idea is worth pursuing, and I’m glad to see work in this direction. The experimental results should benefit the community.

In addition, I agree with the intuition that summation is not necessarily the only / optimal approach for aggregating different token-level rewards, and it’s great to see that the authors have attempted other aggregation functions.




**Weaknesses:**


Reward function retraining seems expensive. Is there an analysis on the compute cost?

I just want to make sure that the authors have ensured that their ROUGE computation is fair.
BART paper and the “Is Reinforcement Learning (Not) for Natural Language Processing: Benchmarks, Baselines, and Building Blocks for Natural Language Policy Optimization” paper have different ROUGE scores for lead-3 baseline, which is a bit weird. BART’s lead-3 result is higher. I just want to make sure in Figure 2, for example, the ROUGE scores are all comparable.

Related: Is there a reason why Table 2 results are run using T5-small instead of a larger model? Would the trend still hold when T5 gets a lot larger?

My understanding is that the reward is Meteor in summarization experiments. Although the summaries are evaluated by ROUGE, what are the Meteor scores (the actual rewards being optimized) on dev/test set? This detail seems to be missing, but apologies if I missed it.


**Questions:**

A few questions are in the "weaknesses" section.

**Limitations:**

I don't see significant discussion on potential limitations. Please let me know if I missed those paragraphs.

---

> ### Author Rebuttal · Authors · 2023-08-06
>
> Thank you for your careful reading of our paper and insightful comments.
> We appreciate it if you can also consider our additional results and clarifications in the **General Response**.
> Below are our detailed responses to your questions.
>
> > **Q1.** The reward-retraining scheme seems expensive.
>
> **A.** We thank the reviewer for raising this important concern.
>
> In our algorithmic design, we reduce the computational cost of the reward-retraining scheme by  only periodically retraining our reward model during the first half of the LM-training process, rather than over the entire training process.
>
> By contrast, as discussed in Appendix G (Line 1007-1017), the baselines RLPrompt [1] and RL4LMs’ (supervised+) PPO/NLPO [2] retrain their value-functions in every optimization step, throughout the whole LM-training process. This can be more demanding and expensive than our method.
>
> Therefore, compared with the baselines, we believe that our method is efficient and thrifty.
>
> [1] Deng, Mingkai, et al. "Rlprompt: Optimizing discrete text prompts with reinforcement learning." arXiv preprint arXiv:2205.12548 (2022).
>
> [2] Ramamurthy, Rajkumar, et al. "Is Reinforcement Learning (Not) for Natural Language Processing?: Benchmarks, Baselines, and Building Blocks for Natural Language Policy Optimization." arXiv preprint arXiv:2210.01241 (2022).
>
> > **Q2.** Make sure that the ROUGE computation is fair.
>
>
> **A.** We thank the reviewers for this careful reminder.
>
> We ensure that our ROUGE computation is fair by following the codebase of RL4LMs [2], to have a fair comparison with our baselines.
>
>
> > **Q3.** Why results in Table 2 are run using T5-small instead of a larger model? Would the performance gain still hold when T5 gets larger?
>
> **A.** As discussed in Line 263, results in Table 2 are run using the standard T5-small model because of our limited computing resources. Furthermore, using T5-small facilitates our comprehensive ablation study in Section 4.3 and Appendix A.2.
>
> We verify the performance of our method under a larger LM by scaling up from T5-small to T5-base in Section 4.2 (Line 285-290), where the model is enlarged by about 3.5 times. The results are shown in Figure 2, with detailed numbers available at Table 4 in Appendix A.1 and Table B in the added PDF. It is clear that our method still performs favorably against the strong baseline results directly cited from RL4LMs [2], when T5 gets a lot larger.
>
> Therefore, we believe that our experiments are sufficient to demonstrate the effectiveness and benefits of our method, based on our main results, ablation studies, and the newly uploaded PDF.
>
>
> > **Q4.** My understanding is that the reward is Meteor in the summarization task. What are the Meteor scores on this task?
>
>
> **A.** The reviewer is correct that in the summarization task, we use the Meteor score to obtain the sequence-level preference, which we ground into token-level guidance for LM training by our proposed method.
>
> The Meteor scores on CNN/DM summarization under T5-base LM can be found at Figure 2 (Section 4.2), with detailed numbers at Table 4 (Appendix A.1) and Table B in the added PDF. By standard, these results are on the test set. The baseline results are directly cited from RL4LMs [2].
>
> It is clear that our method outperforms these strong baselines in the Meteor metric as well.
>
> > **Q5.** No significant discussion on limitations.
>
> **A.** Due to the page limit, we defer the discussion on limitations to Appendix I (Line 1031-1041).

---

> > ### Comment · Reviewer_gmdH · 2023-08-13
> > **Response to authors**
> >
> > Thank you for the rebuttal. I went through the details. Here are a few more comments.
> >
> > Q1 (reward retraining): Thanks for the response. It's great that the authors have taken steps to make reward retraining more efficient. I would appreciate it if there's an analysis on
> > - how much compute is spent on reward retraining vs. other training,
> > - and how much compute the authors' techniques are saving, with respect to regular/vanilla reward retraining.
> >
> > Q3: Good to know that the trends hold on T5-base as well. No need for experiments given that the authors say there is a lack of compute, but do the authors think that the trend will generalize to even larger models?
> >
> > Additionally, I realized that there should be discussion on other weighted MLE approaches in NLP, as I don't think it's an extremely common technique. Just a few examples: RAML in https://arxiv.org/abs/1609.00150, weighted MLE in grammatical error correction for MT in https://arxiv.org/abs/1804.05940, weighted MLE in table-to-text generation in https://aclanthology.org/2021.acl-short.11/, GOLD in https://arxiv.org/abs/2009.07839.
> >
> > Raising my score from 5 to 6, given that I'm satisfied with the other parts of the response.

---

> > > ### Comment · Reviewer_gmdH · 2023-08-14
> > > **Raising score further**
> > >
> > > On second thought, I'm raising the score to 7. I think the token-level reward direction and the different token-level reward aggregation methods (other than summation) are especially interesting.

---

> > > > ### Author Response · Authors · 2023-08-14
> > > > **Response to Reviewer gmdH**
> > > >
> > > > We deeply appreciate your feedback and are delighted that our paper and rebuttal merit your raising the score.
> > > >
> > > > For reward retraining, we will follow your suggestion and rerun our method to carefully time reward retraining v.s. LM training. We will also rerun the direct baselines RLPrompt and RL4LMs to compare our method’s timing with. These results will certainly be added in the next version of our manuscript. Regarding the saved time, in our preliminary study, we observed that our techniques (“retraining the reward model only during the first half of the LM training process”) can save about 25-30% computation, compared to vanilla reward retraining (“retraining the reward model throughout the entire process”).
> > > >
> > > > For the trend, yes, we believe that the trend will continue to hold on even larger models, based on our experiment of scaling up from T5-small to T5-base.
> > > >
> > > > Finally, we are grateful to the reviewer for the great suggestion on related works. As a heads up, RAML uses sequence-level reward (negative edit/Hamming distance), rather than token-level as in our method. We will certainly add a detailed discussion on the weighted-MLE methods in NLP, especially your four suggested papers, in our revised manuscript.

---

### Official Review · Reviewer_xDED · 2023-07-07

**Soundness:** 3 good
**Presentation:** 3 good
**Contribution:** 2 fair
**Rating:** 7
**Confidence:** 4

**Summary:**

The paper proposes to solve the issue of granularity match in preference-based tuning of LMs (RLHF for e.g.,), that is task-based preference is defined at the sequence level (via pairwise preference learning) while the reward model training and policy optimization is done at the token level. The paper proposes an extra step to ground the token-level reward into the sequence-level preference. Evaluation is done on two tasks (prompt generation and summarization) and some improvement is shown compared standard RL-based method such as PPO and NLP as well as vanilla supervised models



**Strengths:**

- The paper is well written and explains the proposed method well and the experimental setup is well documented.
- The studied problem is interesting and relevant especially since RLHF methods are becoming more common nowadays.
- Evaluation is done against strong baselines.
- The proposed method seems to bring some improvements on the two studied tasks

**Weaknesses:**

- **Limited tasks**: Evaluation is only two tasks and it's not clear why these two particular tasks were selected. I imagine other tasks should be used, where preference-based learning is relevant. Such tasks include toxicity avoidance and controllable/constrained generation.
- **No human evaluation**: No human evaluation is done on summarization: We know that RL methods are good at hacking metrics and I would expect human evaluation on the summarization task to support the improvements in ROUGE.
- **Some choices are not justified**: For example, in Algorithm 2, why is the reward model retrained only for the first half of the iterations? Why not for the entire LM training? Why is a max-entropy gradient added to REINFORCE in Eq 5? Did you add the same for the baselines?
- **Not clear where improvement compared to the baselines comes from**: For example in Algorithm 2, the reward model is trained alternatively with the  LM. Are all baselines trained the same way? If not, then there is no way to know if improvement comes from this method of training or from the sequence-level grounding.

**Questions:**

- See last two points in Weaknesses
- In the ablation in section 4.3,  (b), How is this done? you just replace $f(\lbrace r_\phi(s_t^k, a_t^k)\rbrace)$  with $\mathcal{R(s^k_T)}$?


**Limitations:**

I think the authors addressed some of the limitations of their work.

---

> ### Author Rebuttal · Authors · 2023-08-06
>
> Thank you for your time and careful review. We first want to bring your attention to our **General Response** for our human evaluation results on CNN/DM summarization.
> Below are our detailed responses to your other concerns.
>
>
> > **Q1.** Evaluation is only on two tasks. It's not clear why these two tasks were chosen. Some other tasks can be used.
>
> **A.** Due to the page limit and our limited computational resources, in this paper, we test our method on two carefully-chosen, distinct, and representative LM tasks.
>
> Specifically, the prompt-generation task represents few/zero-shot learning since there are no ground-truth prompts. Text summarization represents a standard LM task, where there is a set of supervised samples and the generated sequences are of variable-length and are relatively long. Furthermore, on these two tasks, baseline methods/results within our computational budget are easily accessible.
>
> Moreover, we conduct extensive ablation studies in Section 4.3 and Appendix A.2 to comprehensively demonstrate the efficacy and benefit of our method. Therefore, we believe that our experiments are sufficient to validate our method.
>
> We agree with the reviewer that there are many exciting further applications of our method, such as toxicity avoidance and controllable generation that you suggest. We believe that the (potentially) wide applicability of our method is a merit and we will certainly test on these tasks in our future work.
>
>
> > **Q2.** Why is the reward retraining only conducted in the first half of the iterations? Why not over the entire LM-training process?
>
> **A.** We retrain the reward model only during the first half of the iterations in order to save computational cost. Our preliminary study suggests that retraining the model throughout the entire LM-training process doesn't yield substantial performance gains, making it less computationally worthwhile.
>
>
> > **Q3.** Why do you add a max-entropy gradient to the REINFORCE-style update Eq. (5)? Did you add the same for the baselines?
>
>
> **A.** As discussed in Line 161-163, since we want multiple generated texts in typical LM tasks, we add the max-entropy gradient so as to capture multiple good behavior-modes (good generated texts).
>
> The REINFORCE-style update Eq. (5) is adopted in the prompt-generation task. The baseline method RLPrompt [1] is based on soft Q-learning [2], which is a maximum-entropy Q-learning method for text generation. By its design, it naturally contains a max-entropy gradient. We kindly note that the max-entropy gradient may not be applicable to other baselines due to their specific nature/designs. Their results are directly cited from the literature.
>
> Therefore, we believe that the comparisons between our method and the baselines are fair.
>
> [1] Deng, Mingkai, et al. "Rlprompt: Optimizing discrete text prompts with reinforcement learning." arXiv preprint arXiv:2205.12548 (2022).
>
> [2] Guo, Han, et al. "Efficient (soft) q-learning for text generation with limited good data." arXiv preprint arXiv:2106.07704 (2021).
>
>
> > **Q4.** Not clear if improvement comes from the reward-retraining scheme or from the main contribution: the preference-grounded token-level guidance.
>
> **A.** We respectfully refer the reviewer to our ablation study in Section 4.3 (b), where we remove the reward-function retraining scheme. It is clear that ​​without the reward-retraining scheme, our method still performs competitively against the strong baselines, which confirms the benefit of our preference-grounded guidance and is acknowledged by **Reviewer gmdH**.
>
> As discussed in Appendix G (Line 1007-1017), the reward-retraining scheme does not give our method an unfair advantage over the baseline methods. In particular, the baselines RLPrompt [1] and RL4LMs’ (supervised+) PPO/NLPO [3] retrain their value-functions in every optimization step, which can be more demanding than our method.
>
> Therefore, we believe that our improvement over the baselines should come from our main algorithmic contribution: the preference-grounded token-level guidance, rather than an unfair training scheme.
>
>
> [3] Ramamurthy, Rajkumar, et al. "Is Reinforcement Learning (Not) for Natural Language Processing?: Benchmarks, Baselines, and Building Blocks for Natural Language Policy Optimization." arXiv preprint arXiv:2210.01241 (2022).
>
>
> > **Q5.** How is the ablation in Section 4.3 (b) done?
>
> **A.** In our ablation study in Section 4.3 (b), we remove the reward-function retraining scheme. This can be seen from Algorithm 2 as removing the “if” block containing “Re-train $r_\phi$ by Algo. 1 without re-initialization.” In other words, we train the reward function $r_\phi$ only at the beginning of the LM-training process and fix it thereafter (never retrain it).

---

> > ### Comment · Reviewer_xDED · 2023-08-13
> > **response to rebuttal**
> >
> > I acknowledge reading the author's rebuttal which aimed to address some of my concerns. Here's my response to the author's rebuttal:
> >
> > > We retrain the reward model only during the first half of the iterations in order to save the computational cost.
> >
> > I wonder how much compute can actually be saved by this. I'd still argue that this is a non-justified design choice.
> >
> > > We kindly note that the max-entropy gradient may not be applicable to other baselines due to their specific nature/designs.
> >
> > In this case, I would expect an ablation showing that your approach still outperforms the baselines without the max-entropy gradient. I understand that achieving a 100% fair comparison with the baselines is hard, but the authors add extra layers of complexity through some design choices, making a fair comparison much harder than it needs to be.
> >
> > > We respectfully refer the reviewer to our ablation study in Section 4.3 (b), where we remove the reward-function retraining scheme.
> >
> > Thanks for the reference. I can now see that your approach does not rely on the reward re-training scheme, which begs the question as to why it is a part of your approach since the improvement introduced as a result is extremely minor (0.08 ROUGE-1 points, for e.g.) Again, this points to some design decisions made by the authors that are not fully justified. Respectively, it seems like the proposed approach relies on so many moving parts that it has become unclear where the contribution actually is or whether these extra layers of complexity are justified.
> >
> > Overall, I thank the authors for their rebuttal. However, nothing in the author's response merits raising my score so far.

---

> > > ### Author Response · Authors · 2023-08-13
> > > **Further Response to Reviewer xDED**
> > >
> > > Thank you so much for the quick response.
> > >
> > > We respectfully disagree with your comment that our paper has many unjustified design choices. We address your concerns in details below.
> > >
> > > > **Q1.** How much compute can be saved by only retraining the reward model during the first half of the training process?
> > >
> > > **A.** In our preliminary study, we observed that “retraining the reward model only during the first half of the LM training process” can save about 25-30% computation, compared to “retraining the model throughout the entire LM-training process.”
> > >
> > > Intuitively, “retraining only during the first half” can clearly reduce computation compared to “retraining the model throughout the entire LM-training process.” We believe that the computation-saving nature of this design can sufficiently justify itself. Therefore, we respectfully disagree with the reviewer’s argue that this is an unjustified design choice.
> > >
> > >
> > > > **Q2.** Expect an ablation showing that your approach still outperforms the baselines without the max-entropy gradient.
> > >
> > > **A.** We first kindly remind that as discussed in **Q3** of the previous Rebuttal, our most direct competitor in the prompt task RLPrompt (Line 246-249), has a max-entropy gradient as well.
> > >
> > > As a recap, we test our method on the task of discrete-prompt generation. Apart from RLPrompt, the other discrete-prompt baselines use more demanding designs and requirements than our method (or RLPrompt), such as human handcraft and/or other forms of human efforts.
> > > Specifically,
> > > * “Manual Prompt” uses hand-crafted prompts.
> > > * “In-Context Demo” requires (human) selecting one training example per class.
> > > * “Instructions” requires manually created task descriptions and label definitions.
> > > * “GrIPS” requires instructions designed for humans.
> > >
> > > By contrast, our method does not require any of these demanding human efforts, and while less demanding, can still outperform those methods.
> > >
> > > We clarify that the max-entropy gradient is part of our LM-training objective Eq. (5), which targets the few/zero shot setting, where we do not assume the availability of supervised data, e.g., the ground-truth prompts.
> > >
> > > Due to this challenging setting, we believe that it is unfair for us to remove this component and compare with a baseline having max-entropy gradient (RLPrompt) or with baselines using more demanding designs, such as human handcraft and/or other forms of human efforts (e.g., “Manual Prompt”, “In-Context Demo”, “Instructions”, “GrIPS”).
> > >
> > > Based on the above clarification, we believe that our comparison with the baselines is fair.
> > >
> > > In addition to the above clarification, we kindly note that RLPrompt also has max-entropy gradient, and in its paper compares with the same set of baselines as our paper. Since RLPrompt’s experimental comparison has been perceived as fair by the community, we believe that our comparison with these baselines is fair as well.
> > >
> > >
> > > > **Q3.** Why is the reward-retraining scheme a part of your approach since the introduced improvement is minor on the summarization task?
> > >
> > > **A.** As discussed in Line 314-317 of our main paper, the gain of this scheme depends on the zero-shot ability of the initial LMs. Specifically, in the prompt task where the initial LM has little zero-shot ability, reward-function retraining can particularly be helpful to both improve performance and reduce variance. As a numeric example for Figure 4 in our paper, in the “SST-2” dataset of the prompt task, reward-retraining scheme improves the result from $90 \pm 2.9$ to $92.6 \pm 1.7$.
> > >
> > > Meanwhile, we agree with the reviewer that *on the summarization task*, the reward-retraining scheme indeed may not help the results as much, since the initial LM has some zero-shot ability (Line 317-319).
> > >
> > > Since our paper wants to build a general framework that is applicable to both of these distinct settings, we apply the reward-retraining scheme to both tasks in our main results. Then, the ablation study in Section 4.3 (b) serves as a further experimental explanation of this scheme.
> > >
> > > ***
> > >
> > > Overall, based on this response and the previous Rebuttal, we believe that our method has similar complexity at least with our direct competitors RLPrompt and RL4LMs. Furthermore, we believe that we have sufficiently validated our method’s building blocks through our comprehensive ablation study in Section 4.3 and Appendix A.2, which is acknowledged by Reviewer **9EYo**.
> > >
> > > We hope this additional clarification can address all your concerns and merit raising your score. Please kindly let us know if you have any remaining concerns.

---

### Official Review · Reviewer_wRKD · 2023-07-07

**Soundness:** 2 fair
**Presentation:** 2 fair
**Contribution:** 3 good
**Rating:** 4
**Confidence:** 4

**Summary:**

This paper proposed break the pairwise sequence preference into token-level guidance signal by iterating between learning a token-level reward from sequence level preference and improving LM with the learned token level guidance. Experiments are conducted on two language generation tasks and competitive results are reported.

**Strengths:**

- The paper propose to address a problem of how to effectively ground sequence-level preference into LM finetuning.
- studies different aggregation function to break sequence level preference into token level reward function
- the setup with supervised data to weight token level MLE by token level reward is interesting

**Weaknesses:**

- in general the paper is difficult to follow, e.g. Figure 1 is not self explanatory, experimental setups and datasets could be use more details.

- a core motivation/hypothesis of the paper between sequence and token level losses is questionable. E.g. abstract mentions "a granularity mismatch between the preference and the LM training losses". It is wrong to categorize LM loss as token-level loss, because it is MLE over target sequence on both token level and sequence level. there are many unsupported assumptions around the mismatch of sequence and token level losses in the introduction section.

- in PPO, even though the reward function is sequence level, there is a token-level value network. Discussion and ablation of how that is trained is very relevant to choosing the aggregation of your token level reward model.

- the dataset to test your method is not proper. "simulate the sequence-level preference by the classical Meteor score and report the standard ROUGE scores" is questionable because it is not real sequence level feedback. On the other hand, there are datasets readily available, e.g. "Learning to summarize from human feedback" which contains real human preference data on summarization tasks.

**Questions:**

NA

---

> ### Author Rebuttal · Authors · 2023-08-06
>
> We thank the reviewer for raising several important questions. Please would you first check our **General Response**, which should address your concern on testing against the RLHF summarization paper. We answer your remaining questions in details below.
>
> > **Q1.** The clarity of our paper.
>
> **A.**
> We thank the reviewer for raising this issue. We will carefully revise our paper and include more details into Figure 1 in the next version of our manuscript. Meanwhile, we would like to refer to **Reviewer gmdH**’s Summary section for a good overview of our paper.
>
> We provided details on experimental setups and datasets in Sections 4.1 and 4.2. Due to the page limit, we deferred additional experimental details to Appendix B and additional details about the prompt task to Appendix D.
>
> Finally, we notice that **Reviewer xDED** praised our paper as *“well written … and the experimental setup is well documented”*; while **Reviewer gmdH** thinks that our Figure 1 *”is a good illustration”*. Therefore, we will highly appreciate it if you can point out explicitly the confusing parts of our paper.
>
> > **Q2.** About the validity of our core motivation and categorizing LM loss as a token-level loss.
>
> **A.** We respectfully disagree with the reviewer that the core motivation of our paper is questionable and that it is wrong to categorize LM loss as token-level loss.
>
> As discussed in Line 17, we consider LM (cross-entropy) loss as token-level because each token position has a corresponding term in the overall training loss.
> By contrast, the preference/feedback is sequence-level because there is only one feedback for the entire text-sequence, rather than densely at each intermediate timestep.
>
> Intuitively, the granularity mismatch comes from the fact that the LM needs to decide each token, while the preference/feedback is available only after the entire sequence has been generated and is only at the sequence level.
>
> As discussed in Section 3 (Line 191-199), some prior studies have attempted similar/related problems, but are under more restricted/ideal settings.
>
> Therefore, we believe that our core motivation is valid and sound, which is also supported by the other reviewers.
>
> > **Q3.**  In the introduction section, there are many unsupported assumptions on the mismatch of sequence- and token-level losses.
>
> **A.** We will be deeply appreciated if you can explicitly point out the unsupported assumptions in our introduction section, apart from the above **Q2** that we have clarified.
>
>
> > **Q4.** Despite the sequence-level feedback, there is a token-level value network in PPO. And how that is trained is very relevant to choosing the aggregation of your token-level reward model.
>
> **A.** We agree with the reviewer that there is a token-level value network in PPO. However, the main problem lies in learning this value function.
>
> Specifically, the mismatch between sequence-level feedback and token-level LM training/generation leads to the problem of delayed feedback (sparse reward-signal) when applying RL methods to LM tasks. It is known in the literature [1, 2] that with only sparse rewards, it can be difficult to estimate the token-level value functions in RL methods, including PPO. In particular, [1] points out that the standard value-function learning can suffer from *“the unstable per-step bootstrapping-style training with sparse reward signals.”*
>
> In Appendix F.2, we provide a detailed discussion on this delayed feedback problem in RL-based LM training, which we believe applies to PPO. In Section 4.2 and Table B in the added PDF, we show that our method outperforms the benchmarking results of (Supervised+) PPO cited from RL4LMs [3].
>
> We are confused about the second part of your comment. We are unaware of other ways to train the value network in PPO, apart from the standard one implemented in our baseline RL4LMs [3]. We are also confused about why that will be relevant to choosing the aggregation of our token-level reward model. As a gentle reminder, in our paper, the aggregation function is used to train the token-level reward function $r_\phi$ (Line 104-109).
>
> We will highly appreciate it if you can provide more details on your comment so that we can respond better, rather than only taking an educated guess.
>
> [1] Guo, Han, et al. "Efficient (soft) q-learning for text generation with limited good data." arXiv preprint arXiv:2106.07704 (2021).
>
> [2] Sutton, Richard S., and Andrew G. Barto. Reinforcement learning: An introduction. MIT press, 2018.
>
> [3] Ramamurthy, Rajkumar, et al. "Is Reinforcement Learning (Not) for Natural Language Processing?: Benchmarks, Baselines, and Building Blocks for Natural Language Policy Optimization." arXiv preprint arXiv:2210.01241 (2022).
>
> > **Q5.** Simulating the sequence-level preference by the Meteor score is questionable because it is not real sequence-level feedback.
>
> **A.** We respectfully disagree with the reviewer. Since the Meteor metric scores the entire generated text-sequence, it provides a sequence-level feedback on LM generation.
> Therefore, we believe that it is valid to use Meteor to simulate the sequence-level preference, i.e., the higher score the better generated sequence, as discussed in Line 101-103. Hence, our summarization experiments on datasets CNN/DM and XSum are proper.
>
> In general, we believe that automatic evaluation metrics can be a valid and rich source of sequence-level preference, apart from real humans.

---

> > ### Comment · Reviewer_wRKD · 2023-08-17
> >
> > Thank you authors for the responses. I increased my scores after reading your responses and other reviewer's comments.
> >
> > Q2: It is the autoregressive factoring of the LM that ties sequence probability to token probability.
> > Viewing LM cross entropy loss as token-level loss is one view of the problem.
> > An equivalent view is that,  if we conduct MLE on sequence probability, which is autoregressively factored to the product of token probabilities, seq_prob = \pi_i token_prob_i, we would get the exact same loss function as summation of token level MLE.
> > Using this view, the cross entropy loss can be explained without mention of token-level losses, and also from data perspective, the SFT data we collected are full sequences (one argument could be that we do not collect token/prefixes but always collect full sequences).
> > Equivalently, any signals that are collected on the sequence level are propagated to token-level because of the autoregressive factoring.
> > I acknowledge that sequence level losses and token level losses are often mentioned in literature, however in my opinion, the there is no mathematically difference on sequence-level and token-level losses for autoregressive LMs.

---

> > > ### Author Response · Authors · 2023-08-17
> > > **Further response to Reviewer wRKD**
> > >
> > > Dear Reviewer wRKD,
> > >
> > > Thank you so much for your response and your raising the scores.
> > >
> > > We agree with the reviewer that the token probabilities in LM have connection with the sequence probability. But, as per you commented, *”Viewing LM cross entropy loss as token-level loss is one view of the problem”*. Therefore, we believe that our core motivation is solid, and it is valid to categorize LM loss as token-level loss. We agree that there are other views of the same problem, but we do not preclude those other views and we do not aim at drawing a dichotomy. We will add a clarification on this in our revised manuscript.
> > >
> > > We would like to reiterate that the goal of this paper is to ground sequence-level feedback into (dense) token-level guidance for LM training (Section 1). In our experiments (Section 4), we demonstrate that our viewpoint and method can be beneficial on two distinct representative LM tasks/settings.
> > >
> > > Specifically, with supervised data, even though the classical supervised MLE can work, we show in Section 4.2 that our token-level reward-weighted MLE can perform better. Intuitively, the reward/guidance-weighting scheme emphasizes the important tokens in the supervised sequences and downweights the unimportant ones, and therefore can better utilize the LM capacity and the optimization budget (Line 167-171).
> > >
> > > In the challenging setting of without supervised data, the LM needs to discover each token by itself. We show in Section ​​4.1 that our dense token-level guidance can improve the performance/quality of the generated sequences.
> > >
> > > In both settings, we experimentally show that our dense token-level guidance can be more effective for LM training than the delayed/ungrounded *native* sequence-level feedback (Line 246-249 and Line 276-278).
> > >
> > > ***
> > >
> > > We wish this discussion can clarify the contribution of our paper and merit raising your rating further. Please kindly let us know if you have any remaining concerns.

---

### Official Review · Reviewer_XL7b · 2023-07-07

**Soundness:** 3 good
**Presentation:** 3 good
**Contribution:** 2 fair
**Rating:** 5
**Confidence:** 4

**Summary:**

The paper presents a new approach to training language models that address the mismatch between coarse-grained sequence-level preferences and fine-grained token-level rewards. With more fine-grained rewards, the proposed framework reduces the reliance on supervised data. Specifically, given the preference of a set of sequences, it first trains a reward function such that the aggregated rewards over tokens can satisfy the preference. Then, it applies REINFORCE to update the language model. The framework is evaluated on two text generation tasks: generating text prompts for few-shot text classification and text summarization. Compared with baselines, the proposed method improves the performance on both tasks.

**Strengths:**

Revisiting the mismatch between sequence-level and token-level feedback in fine-tuning language models (LLMs) is of particular interest to me. In scenarios where supervised data for a target domain downstream task is scarce, and fine-tuning LLMs is computationally expensive, exploring data-efficient methods becomes important. The proposed method is straightforward and reasonable to me.

**Weaknesses:**

1. Using discrete-prompt generation to evaluate the proposed method is not convincing. The chosen task itself lacks meaningfulness, and a better way to achieve good performance may be self-training as introduced in [1]. I can understand the goal of the experiment is to evaluate the quality of the generated sequence, but there are other more reasonable generation tasks such as personalized chatbot.
2. The performance of the proposed method of text summarization falls significantly short of the SOTA, which leaves me unconvinced by the results. For instance, the ROUGE-L score for the current SOTA on CNN/DailyMail has surpassed 44, and for XSum it stands at 40.4 [2]. In contrast, the proposed method achieves scores of 38.17 and 26.33 respectively. For CNN/DailyMail, the results are only slightly better than the trivial lead-3 baseline. Considering that the proposed method is complex, I expect more improvement. I can understand the low performance may be partially due to model size. However, without using the model size of a state-of-the-art model, it’s hard to evaluate the effectiveness of the proposed method because It’s easier to boost the performance on a small model. In [3], they challenge the effectiveness of RL and argue that using a few human-curated prompts and responses is more effective for finetuning LLMs.
3. Using ROUGE as the only evaluation metric for summarization is not convincing. At least human evaluation or language model scores such as BERT-score/BART-score should be used. Also, is it possible to obtain preference via language model scores?
4. The proposed method lacks novelty. Previous studies have extensively discussed the mismatch between token-level and sequence-level rewards. Using a token-level reward function to estimate the reward for each token is not a new idea. Several related works are missing [4][5].

[1]ExploitingClozeQuestionsforFewShotTextClassificationandNatural LanguageInference, Schick et. al.
[2]BRIO:Bringing Order to Abstractive Summarization, Liu et. al.
[3]LIMA: Less Is More for Alignment, Zhou et. al.
[4]SeqGAN: Sequence Generative Adversarial Nets with Policy Gradient, Yu et. al, 2016.
[5]Unsupervised Text Style Transfer using Language Models as Discriminators, Yang et. al.


**Questions:**

1. Are there any generated samples for the discrete prompts for zero-shot classification? Qualitative analysis may make the method more convincing.
2. How does the choice of reward function impact performance? Does using different pre-trained models as reward functions significantly impact the performance?
3. Why the summarization results are significantly below SOTA results? The paper mentions it uses the full training set for finetuning the model, so I expect the performance should at least not be far from SOTA results.
4. How to obtain the preference for discrete prompts of zero-shot classification?

**Limitations:**

Please refer to weaknesses and questions.

---

> ### Author Rebuttal · Authors · 2023-08-06
>
> We deeply appreciate the reviewer for the careful comments.
> We would like to draw your attention to **General Response** for additional results and common responses. The other questions are answered in detail below.
>
> > **Q1.** About the meaningfulness of discrete-prompt generation task. May test on other tasks.
>
> **A.** We respectfully disagree your comment that *“the chosen task (discrete-prompt generation) itself lacks meaningfulness.”* We believe that while the self-training in [1] may give good results, it doesn’t preclude other methods to this problem, such as the recent RL-style method [6] and the recent baselines therein.
>
> We kindly argue that this task is meaningful at least in better human understandability of the generated prompts, compared to soft prompts. We discuss related work in Appendix E (L857-864).
>
> In this paper, we use discrete-prompt generation to test the applicability of our method to the challenging setting of no supervised examples, i.e.,  no ground-truth prompts.
> As discussed in L191-199, this setting makes it infeasible to apply many prior works on learning token-level guidance for LM training, since they typically require abundant supervised examples.
>
> In Section 4.1, we show that our preference-grounded token-level guidance is not only applicable to this task, but can be more effective than many strong recent methods, such as RLPrompt [6].
>
> As discussed in L342, we agree with you that dialog systems can be a further application of our method. But we believe that discrete-prompt generation is an equally reasonable and important task.
>
> [6] Deng, Mingkai, et al. "Rlprompt: Optimizing discrete text prompts with reinforcement learning." arXiv preprint arXiv:2205.12548 (2022).
>
> > **Q2.** LIMA [3] challenges the effectiveness of RL for finetuning LLMs.
>
> **A.** We want to clarify that our method is not a RL method, though some related works and baselines are.
>
> We thank the reviewer for raising LIMA [3]. However, LIMA was submitted to arXiv on 18 May 2023, which is after the NeurIPS submission deadline (17 May 2023). Thus, a discussion on LIMA is out of the scope.
>
> > **Q3.** Is it possible to obtain preference via language model scores?
>
> **A.** Yes, it is possible, as long as LM scores can desirably reflect sequence quality in the specific task.
> As discussed in L101-102, we make no assumption on the preference source.
>
> In our summarization task, we obtain preference by Meteor, to have a fair comparison with the benchmarks in Table 17 of RL4LMs [7] and to avoid overfitting the ROUGE evaluation metrics (L258-259).
>
> [7] Ramamurthy, Rajkumar, et al. "Is Reinforcement Learning (Not) for Natural Language Processing?: Benchmarks, Baselines, and Building Blocks for Natural Language Policy Optimization." arXiv preprint arXiv:2210.01241 (2022).
>
> > **Q4.** About lacking novelty and prior works on using token-level rewards, such as [4, 5].
>
> **A.** We thank the reviewer for suggesting [4, 5]. We will certainly include them in our revised draft.
>
> We agree that some prior works have attempted to learn token-level guidance for LM training. But as we discussed in Section 3 (L191-205), those prior works typically require abundant expert data, making them infeasible for the few/zero-shot settings, such as discrete-prompt generation, where there are no ground-truth prompts.
>
> The suggested related works SeqGAN [4] and Unsupervised Text Style Transfer [5] both fall into this category. Specifically, [4] requires a set of “real sequence data” to pre-train the generator by MLE at the beginning, and [5] needs real target-domain sentences to train LM discriminators. Thus, both [4, 5] are infeasible to our few/zero-shot prompt task.
> Further, similar to our related work “Lin et al. [64]” (L195), the intermediate guidance in [4] comes from Monte Carlo search, which can have high variance and be compute-demanding.
> In [5], the token-level feedback is essentially the token-level probability from the target-domain LM, which can be less general/flexible than our preference-grounded guidance.
> Apart from these, both [4, 5] do not consider the preference relation among multiple generated sequences.
>
> By contrast, our method grounds the sequence-level preference into task-specific token-level guidance for LM training; and is suitable for both standard LM tasks and the low-data regime.
>
> > **Q5.** Samples of the generated discrete prompts?
>
> **A.** Some samples of the generated discrete prompt are in Table 3 (Appendix A.1). A brief discussion is on L250-254.
>
> > **Q6.** How does the choice of reward function impact performance?
>
> **A.** In Appendix A.2 (c) (L740-757), we simulate the preference on the summarization task by two other automatic metrics “Rouge-avg” and “Rouge-avg2”, rather than the Meteor in Section 4.2. We show that the efficacy of our method is not tied to using Meteor as the preference source.
>
> We appreciate it if you could further elaborate this question: *“Does using different pre-trained models as reward functions significantly impact the performance?”* Overall, we think that the answer depends on how different pre-trained models are used as reward functions, and how the performance is measured.
>
> We note that in summarization, our preference is from Meteor or Rouge-style scores. Both we and the results in RL4LMs [7] do not use pre-trained models as reward functions. Thus, a discussion of that may be beyond scope.
> In the prompt task, we obtain the preference by the stepwise metric in RLPrompt [6] (L226-227), to ensure a fair comparison with RLPrompt and the baselines in its paper. We are unsure how different pre-trained models can be used or if changing this preference-obtaining metric will still make fair comparisons.
>
> > **Q7.** How do you obtain the preference for discrete prompts?
>
> **A.** As discussed in L224-225, it’s obtained by the stepwise metric in RLPrompt [6], i.e., the higher the metric value the better. Details of this metric are in our Appendix D (L842-851).

---

### Author Rebuttal · Authors · 2023-08-06

## General Response
We thank all reviewers for the valuable comments. Below are our additional results and responses to common concerns.

> **Q1.** Human evaluation results on the CNN/DM summarization.

**A.**  We conduct human evaluation on the quality of the generated summaries on the CNN/DM dataset under the T5-base LM. We generally adopt the protocol in [1] to evaluate the overall summary quality.
Our model is compared with the baselines Supervised, Supervised+PPO, and Supervised+NLPO in RL4LMs [2]. We also present the result of the reference summaries, which is intended for sanity check rather than method comparison.

Specifically, we randomly picked 100 articles in the test split of CNN/DM and showed to 20 qualified evaluators the summaries generated from each method, along with the article. The method names were anonymized. The evaluators were asked to read the article and score each summary.

Table A in the added PDF shows the average human ratings. We see that our method outperforms all baseline methods Supervised and Supervised+PPO/NLPO. This aligns with our results in Section 4.2 and **Reviewer xDED**’s expectation that human evaluation on the summarization task supports the improvements in ROUGE scores by our method.

[1] Stiennon, Nisan, et al. "Learning to summarize with human feedback." Advances in Neural Information Processing Systems 33 (2020): 3008-3021.

[2] Ramamurthy, Rajkumar, et al. "Is Reinforcement Learning (Not) for Natural Language Processing?: Benchmarks, Baselines, and Building Blocks for Natural Language Policy Optimization." arXiv preprint arXiv:2210.01241 (2022).

> **Q2.** Results on an additional evaluation metric: BERTScore [3].

**A.** As suggested by **Reviewer XL7b**, we add an additional metric BERTScore, which can *“correlates better with human judgments and provides stronger model selection performance”* [3].

Table B in the added PDF shows the results expanded from Figure 2 (Section 4.2) and Table 4 (Appendix A.1).
It is clear that on BERTScore, our method again relatively significantly outperforms the strong baselines from RL4LMs [2].

[3] Zhang, Tianyi, et al. "Bertscore: Evaluating text generation with bert." arXiv preprint arXiv:1904.09675 (2019).

> **Q3.** Test against the task/models/results in the paper “Learning to summarize with human feedback” [1].

**A.** In the review, **Reviewer wRKD** suggests us to test our method against the RLHF summarization paper [1].

The main reason for not using this task/dataset is *evaluation*. Specifically, [1] mostly uses large-scale human study to evaluate the models. This requires hiring a large number of human evaluators, which is beyond our budget and scope.

As shown in Section 4 of [1], the (baseline) models in [1] are of sizes ranging from 1.3 billion parameters to 12.9 billion, which are beyond our computational budget for making a fair comparison. In fact, these model sizes are even much larger than the T5-base model used in the RL4LMs paper [2] (220 million parameters).

We are unaware of benchmarking results on this task/dataset on standard automatic metrics, evaluating a wider variety of algorithms and (smaller) LMs. The limited benchmarks further complicates testing our method on this task.

We agree that it is important to test our method using human feedback. We will certainly conduct this study once resources are ready.

> **Q4.** Why are our summarization results below the SOTA?

**A.** We clarify that the goal of this paper is not chasing the SOTA ROUGE-scores on the summarization task. Rather, our goal is to ground sequence-level feedback into token-level guidance for LM training (Section 1). Therefore, we believe that our direct baselines are methods using ungrounded feedback but the same LM backbone, such as RLPrompt and RL4LMs, which we have carefully compared in our experiments.

Further, apart from summarization, we also validate our method on the task of discrete-prompt generation. We believe that our experiments can clearly show the improvement of our method on existing architectures.

**Reviewer 9EYo** mentions that a BART-Large model can give a higher ROUGE score. However, BART-Large is ~7 times the size of our T5-small backbone, and ~2 times of our T5-base. We kindly argue that it is unfair to compare our results with other methods that use a much larger backbone.

In Section 4.2 (Line 285-290), we provide the results on CNN/DM when scaling up our LM from T5-small to T5-base. It is clear that scaling up the LM size further improves our results, e.g., ROUGE-1 increases from 40.9 to 43.1 while ROUGE-L from 38.2 to 40.0. Detailed results of our method under T5-base LM are in Table 4 (Appendix A.1) or Table B in the added PDF. It is also clear that under a T5-base backbone, our method performs more-significantly better than the strong Lead-3 baseline on CNN/DM.

We respectfully argue that apart from using a much larger LM backbone, SOTA summarization methods can contain some specifically-designed techniques, which may interfere with our method and make it hard to separate out the gain of our method.
For example, the BRIO [4] mentioned by **Reviewer XL7b** proposes a training paradigm that assumes a non-deterministic target distribution, which may not easily integrate into the general paradigm of preference-based LM training that our paper considers.

We will appreciate it if our results can be compared against summarization baselines that are more fair and relevant, such as those in the RL4LMs paper. Meanwhile, we appreciate that **Reviewer xDED** considers our evaluation as *“done against strong baselines;”* and **Reviewer gmdH** thinks our experimental results *“should benefit the community.”*

Finally, we are unsure about **Reviewer XL7b**’s comment that “it’s easier to boost the performance on a small model.” We would love to know details/references.

[4] Liu, Yixin, et al. "BRIO: Bringing order to abstractive summarization." arXiv preprint arXiv:2203.16804 (2022).

---

### Decision · Program_Chairs · 2023-09-21

**Decision:**

Accept (poster)

**Comment:**

The paper presents a novel approach to training language models by addressing the granularity mismatch between sequence-level preferences and token-level rewards. Through iterative training, the method integrates sequence-level preference into token-level guidance. The approach is evaluated on two tasks, discrete-prompt generation and text summarization.

Although experiments are not done at a larger scale with bigger models, reviewers agree that the proposed approach is reasonable and most of the experiments are well executed and compared. The exploration of different aggregation methods for token-level rewards is interesting.